# Fluid Evolution and Ore Genesis of the Qibaoshan Polymetallic Ore Field, Shandong Province, China: Constraints from Fluid Inclusions and H–O–S Isotopic Compositions

**Guang-Yuan Yu [1], Shun-Da Li [1,2,*], Yi-Cun Wang [3] and Ke-Yong Wang [1]**

[1] College of Earth Sciences, Jilin University, Changchun 130061, China; yugy14@mails.jlu.edu.cn (G.-Y.Y.); wkyong@163.com (K.-Y.W.)

[2] College of Geology and Mining Engineering, Xinjiang University, Urumqi 830047, China

[3] College of Resources, Hebei Geology University, Shijiazhuang 050031, China; wangyc15@yeah.net

**\*** Correspondence: sdli16@mails.jlu.edu.cn

**Abstract:** The Qibaoshan polymetallic ore field is located in the Wulian area, Shandong Province, China. Four ore deposits occur in this ore field: the Jinxiantou Au–Cu, Changgou Cu–Pb–Zn, Xingshanyu Pb–Zn, and Hongshigang Pb–Zn deposits. In the Jinxiantou deposit, three paragenetic stages were identified: quartz–pyrite–specularite–gold (Stage 1), quartz–pyrite–chalcopyrite (Stage 2), and quartz–calcite–pyrite (Stage 3). Liquid-rich aqueous (LV type), vapor-rich aqueous (V type), and halite-bearing (S type) fluid inclusions (FIs) are present in the quartz from stages 1–3. Microthermometry indicates that the initial ore-forming fluids had temperatures of 351–397 °C and salinities of 42.9–45.8 mas. % NaCl equivalent. The measured hydrogen and calculated oxygen isotopic data for fluid inclusion water ($\delta^{18}O_{FI}$ = 11.1 to 12.3‰; $\delta D_{FI}$ = −106.3 to −88.6‰) indicates that the ore-forming fluids were derived from magmatic water; then, they were mixed with meteoric water. In the Changgou deposit, three paragenetic stages were identified: quartz–pyrite–specularite (Stage 1), quartz–pyrite–chalcopyrite (Stage 2), and quartz–galena–sphalerite (Stage 3). LV, V, and S-type FIs are present in the quartz from stages 1–3. Microthermometry indicates that the initial ore-forming fluids had temperatures of 286–328 °C and salinities of 36.7–40.2 mas. % NaCl equivalent. The measured hydrogen and calculated oxygen isotopic data for fluid inclusion water ($\delta D_{FI}$ = −115.6 to −101.2‰; $\delta^{18}O_{FI}$ = 12.2 to 13.4‰) indicates that the ore-forming fluids were derived from magmatic water mixed with meteoric water. The characteristics of the Xingshanyu and Hongshigang deposits are similar. Two paragenetic stages were identified in these two deposits: quartz–galena–sphalerite (Stage 1) and quartz–calcite–poor sulfide (Stage 2). Only LV-type FIs are present in the quartz in stages 1–2. The ore-forming fluids had temperatures of 155–289 °C and salinities of 5.6–10.5 mas. % NaCl equivalent. The measured hydrogen and calculated oxygen isotopic data for fluid inclusion water ($\delta D_{FI}$ = −109.8 to −100.2‰; $\delta^{18}O_{FI}$ = 10.2 to 12.1‰) indicates that the ore-forming fluids were derived from circulating meteoric waters. The sulfur isotopes ($\delta^{34}S_{sulfide}$ = 0.6 to 4.3‰) of the four deposits are similar, indicating a magmatic source for the sulfur with minor contributions from the wall rocks. The ore field underwent at least two phases of mineralization according to the chronology results of previous studies. Based on the mineral assemblage and fluid characteristics, we suggest that the late Pb–Zn mineralization was superimposed on the early Cu (–Au) mineralizaton in the Changgou deposit.

**Keywords:** stable isotopes; fluid inclusions; Qibaoshan polymetallic ore field; Jiaodong Peninsula

## 1. Introduction

The Qibaoshan area, located to the northwest of Wulian town in Shandong Province, is an important polymetallic ore field. Four medium-sized ore deposits have been discovered so far: the Jinxiantou Au–Cu deposit, the Changgou Cu–Pb–Zn deposit, the Hongshigang Pb–Zn deposit, and the Xingshanyu Pb–Zn deposit. There have been no breakthroughs in prospecting in this area since the Jinxiantou Au–Cu deposit was discovered in 1978. In 2008, the Institute of Geology and Mineral Exploration of Shandong Province discovered the other polymetallic deposits. These four deposits have proven reserves of 11.2 t of Au, 46,000 t of Cu, 1921 t of Pb, and 492 t of Zn.

Numerous studies have been conducted on the Jinxiantou Au–Cu deposit [1–5]. However, the nature of the ore-forming fluids and the genesis of the deposit are still debated. Wang et al. [1] report that the composition of the gas phase of the fluid inclusions (FIs) contain $CO_2$, and the ore-forming temperature is 150–320 °C. The gold mineralization may related to the $CO_2$ content. Wang et al. [6] proposed that the Jinxiantou deposit is a intermediate–low temperature hydrothermal deposit and the temperature and salinity of the main mineralization stage are 120–290 °C and 2.9–14.7% NaCl equivalent, respectively, based on a study of the FIs from the quartz cement of the breccia-hosted ores. Sun et al. [7] argued that FIs with high temperatures (374–404 °C) and salinities (~48% NaCl equivalent) have been found in quartz granules from the ore-bearing altered porphyry. In addition, the Qibaoshan area has the potential to form a porphyry deposit. Unfortunately, the Pb–Zn polymetallic deposits have received less attention. Previous studies of these polymetallic deposits only described the geological characteristics, the host rock features, and the mineral paragenesis [8,9]. The nature of the ore-forming fluids has not been investigated in detail. More data is required to characterize the mineralizing fluids in order to understand the ore genesis and determine the relationship between the different deposits in the Qibaoshan ore field.

Based on our fieldwork, representative quartz and sulfide samples from the ore veins were selected for analysis. The fluid inclusion (FI) petrography and microthermometry provided data that allowed us to determine the phase ratios, constituents, and trapping temperatures of the ore-forming fluids. The new H–O–S isotopic data that we collected enabled us to determine the origins and evolution of the ore-forming fluids and propose a genetic model for the Qibaoshan polymetallic ore field.

## 2. Regional Geology

The Qibaoshan ore field (Figure 1) is located on the southern edge of the Jiaodong Peninsula, the western side of which is bounded by the Tan-Lu fault and the eastern side of which is bounded by the Sulu high–ultra-high pressure orogenic belt. Subduction of the Yangtze Craton beneath the North China Craton was followed by collision between these two plates during the Triassic, which caused the formation of the Sulu–Dabie ultra-high pressure (UHP) metamorphic belt. The Sulu orogenic belt is the eastern extension of the Dabie orogenic belt, which was later faulted by the left-lateral translation of the Tan-Lu fault (Figure 1A). This area entered the Pacific tectonic domain in the Late Mesozoic. Tectonic activity in the Qibaoshan area triggered intense magmatic hydrothermal events, which provided suitable mineralization conditions. Numerous NE-trending and EW-trending fractures were simultaneously formed during this period.

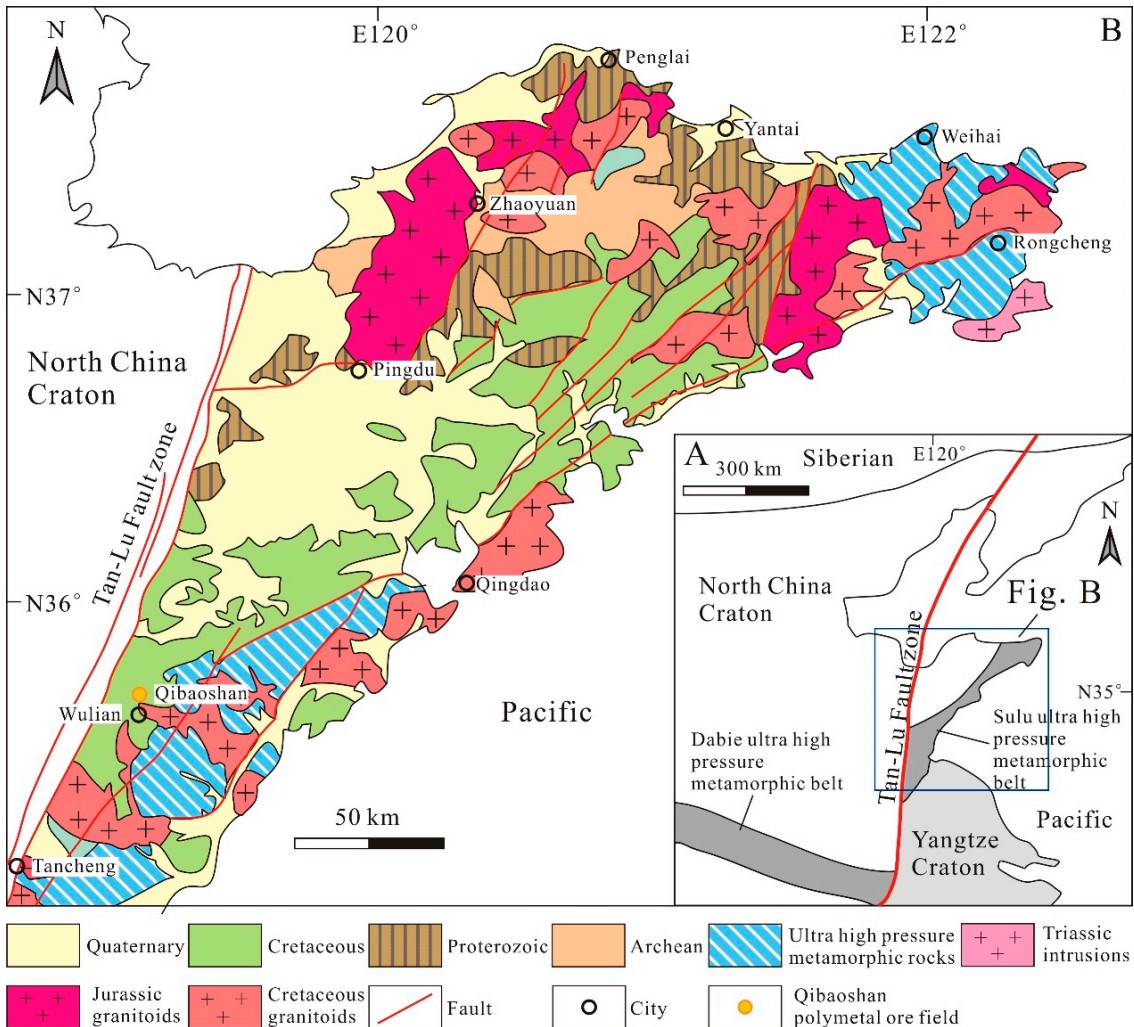

**Figure 1.** (**A**) Tectonic map of eastern China. (**B**) Regional geologic map of the Jiaodong Peninsula (Reproduced with permission from Lan et al. [10]).

The exposed rocks in the area include an assemblage of Archean, Proterozoic, Triassic ultra-high pressure metamorphic, Cretaceous, and Quaternary units (Figure 1B). The Archean rocks include granulite, amphibolite, and marine carbonate, while the Proterozoic rocks include schist, amphibolite, and graphite schist. The Triassic ultra-high pressure metamorphic rocks include granitic gneiss, eclogite, and marble. The Cretaceous rocks include lacustrine sedimentary and continental silicic volcanic rocks. All of these rock units are partially covered by unconsolidated Quaternary sediments.

Mesozoic intrusions are widespread throughout the Jiaodong Peninsula (Figure 1B). The Late Triassic intrusions have U–Pb ages of 225 to 205 Ma [11–13] and are mainly quartz syenite, pyroxene syenite, and alkaline gabbro. These plutonic igneous rocks exhibit typical mantle-derived features and were generated following the collision between the North China Craton and the Yangtze Craton [14,15]. The Late Jurassic granitoids have U–Pb ages of 160 to 150 Ma [16–18] and consist of metaluminous to slightly peraluminous biotite granite, granodiorite, and monzonite, which were probably derived from the partial melting of a thickened Archean lower crust. The Early Cretaceous granitoids have U–Pb ages of 130 to 105 Ma [12,17,19] and consist of granodiorite, porphyritic granite, and monzonitic granite, indicating a mixed source of crustal and mantle components [16,20,21].

## 3. Deposit Geology

The Qibaoshan ore field is located 15 km from Wulian County within a volcanic complex. The exposed area of the complex is approximately 25 km², and the planar shape is an ellipse with a long north–south/southeast axis. The major axis is approximately 5.5 km long, and the minor axis is approximately 4.5 km long. This complex formed as the result of multiple magmatic intrusions (Figure 2).

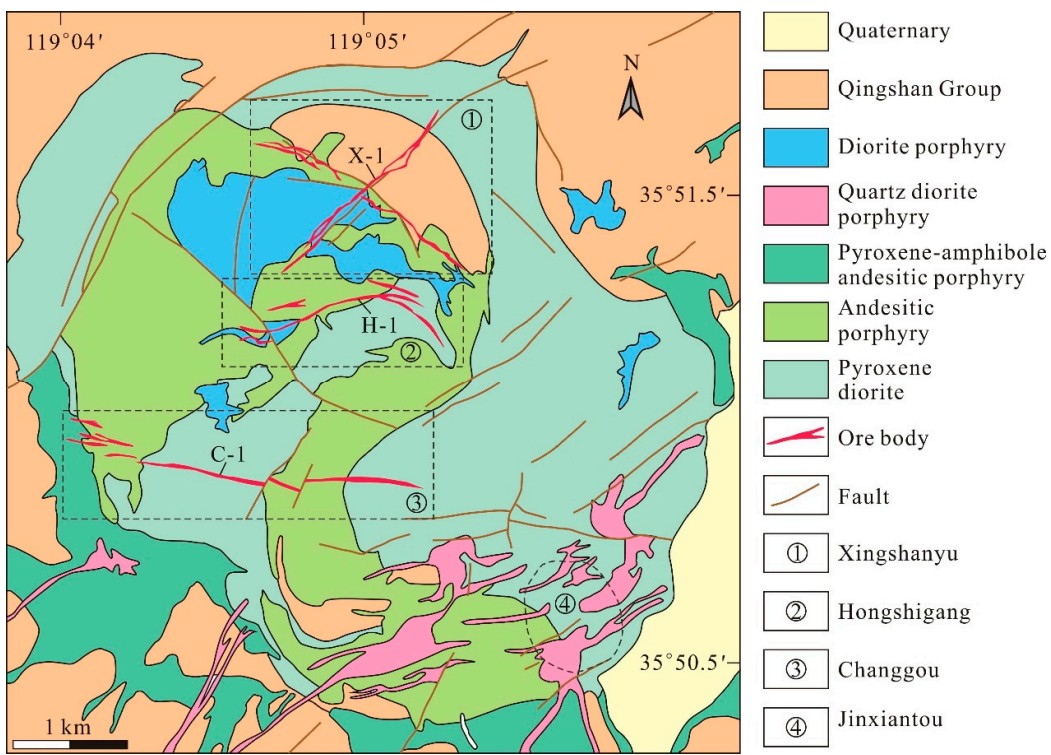

**Figure 2.** Geologic map of the Qibaoshan polymetallic ore field (Reproduced with permission from Zhang [22]).

Based on the emplacement relationships, lithologic characteristics (Figure 3), and U–Pb ages of the complex [23], four phases of igneous rocks were identified: pyroxene diorite (175 Ma), andesite (130 Ma), quartz diorite porphyry (125 Ma), and diorite porphyry (112 Ma). The Qingshan Group pyroclastic rocks have a U–Pb age of 116–117 Ma [24] and are exposed outside of the complex. The Qingshan Group consists of andesite tuffs, tuff breccias, and andesitic brecciated lavas. The strata strike 43° N to 55° E and dip 10°–30° NW. All of these rock units are partially covered by Quaternary clay and grit. Based on their strikes, the faults can be divided into three groups: NE–SW, NW–SE, and E–W trending faults. The ore bodies are mainly present within these faults.

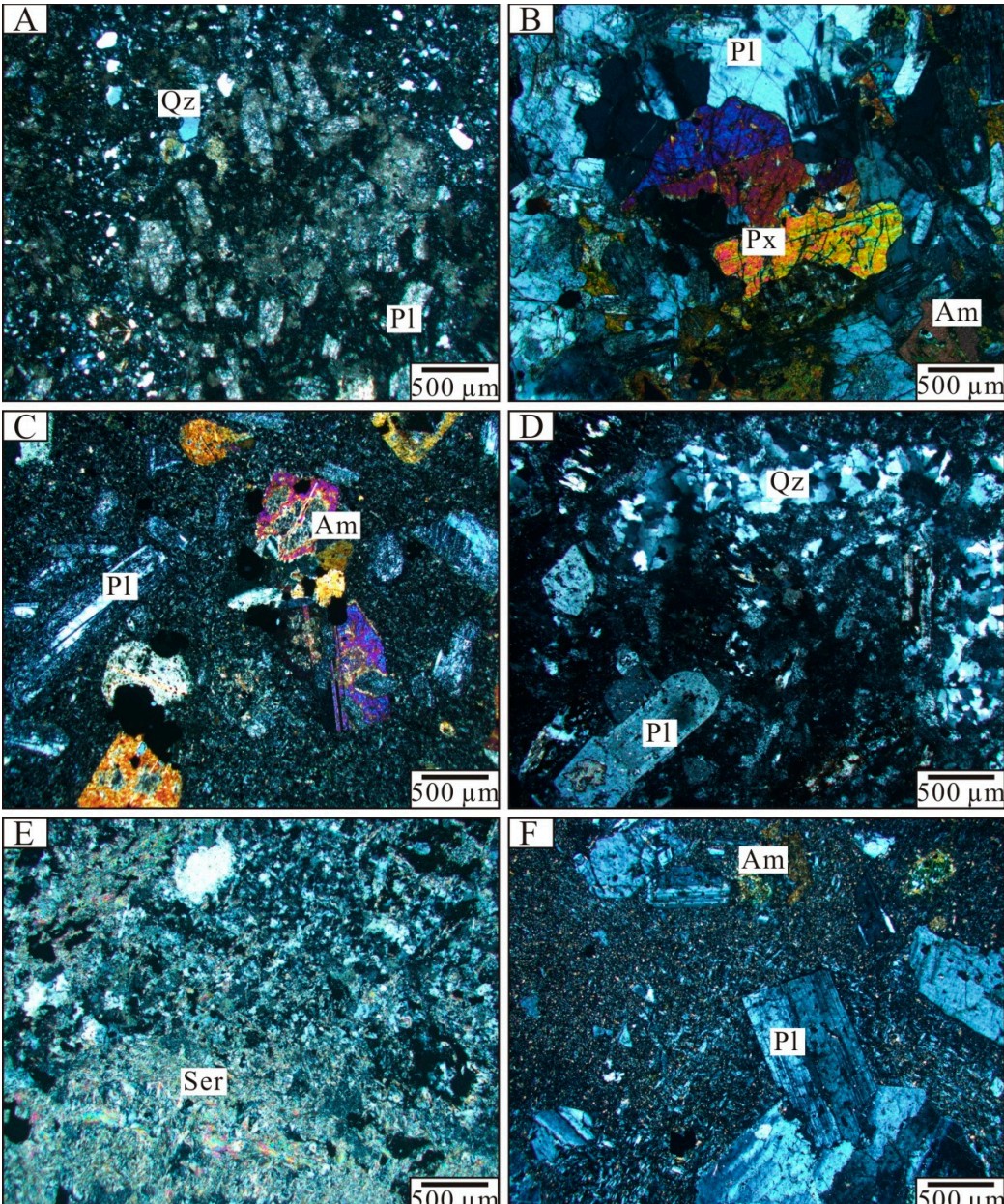

**Figure 3.** (**A**) Photomicrograph of the pyroclastic rocks of the Qingshan Group; (**B**) Photomicrograph of the pyroxene diorite; (**C**) Photomicrograph of the andesitic porphyry; (**D**) Photomicrograph of the quartz diorite porphyry; (**E**) Photomicrograph of the sericitized quartz diorite porphyry; and (**F**) Photomicrograph of the diorite porphyry. Abbreviations: Qz (quartz); Pl (plagioclase); Px (pyroxene); Am (amphibole); Ser (sericite).

The Qibaoshan polymetallic ore field consists of the Jinxiantou, Changgou, Hongshigang, and Xingshanyu deposits (Figure 2). Each ore deposit has unique mineralization characteristics.

## 4. Mineralization Characteristics

### 4.1. Jinxiantou Deposit

Exploration of the Jinxiantou deposit revealed 45 ore bodies (Figure 4A). The ore bodies occur within the quartz diorite porphyry and the adjacent pyroxene diorite. They are mainly controlled by

a cryptoexplosion breccia pipe. The breccia pipe is oval in plane view and quadrilateral in profile. It is 750–800 m long and 250–300 m wide at the surface, and it dips 60°–70° SE (Figure 4A).

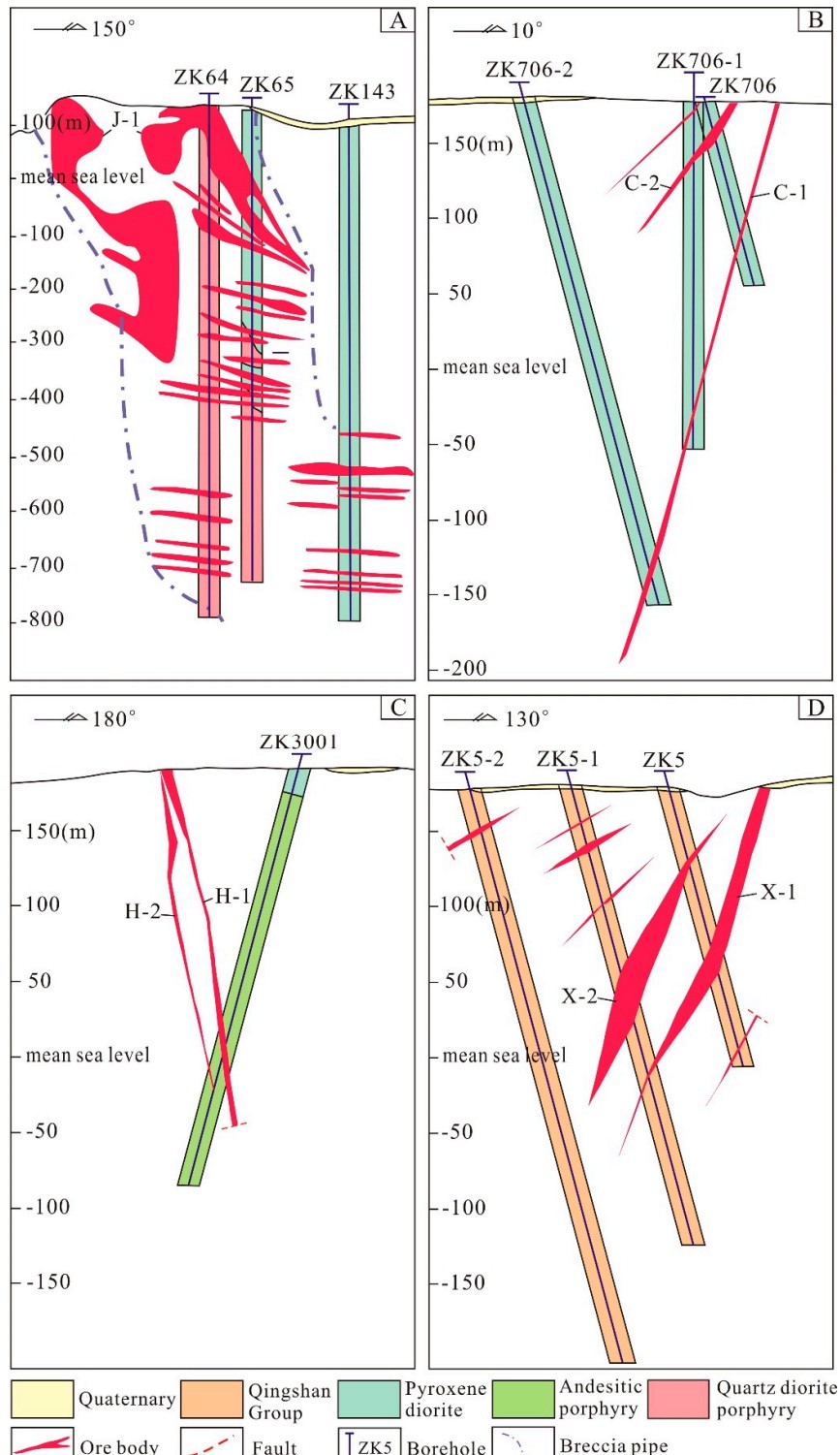

**Figure 4.** Geologic cross-section of the ore bodies (Reproduced with permission from Zhang [22]). (**A**) Jinxiantou deposit, (**B**) Changgou deposit, (**C**) Hongshigang deposit, and (**D**) Xingshanyu deposit (horizontal scale is equal to vertical scale).

The J-1 ore body is the largest one in the deposit. It is 350 m long, and has an average thickness of 14 m (Figure 4A). It strikes NE–SW and dips 45° SE (Figure 4A). The average ore grades are 2.89 g/t Au and 0.96 wt. % Cu.

The assemblage of metal–bearing minerals includes pyrite, specularite, chalcopyrite, minor bornite, digenite, gold, and siderite (Figures 5 and 6). The gangue minerals include quartz, calcite, epidote, and barite. The wall rock alteration is intense and consists of potassic alteration, silicification, and sericitization.

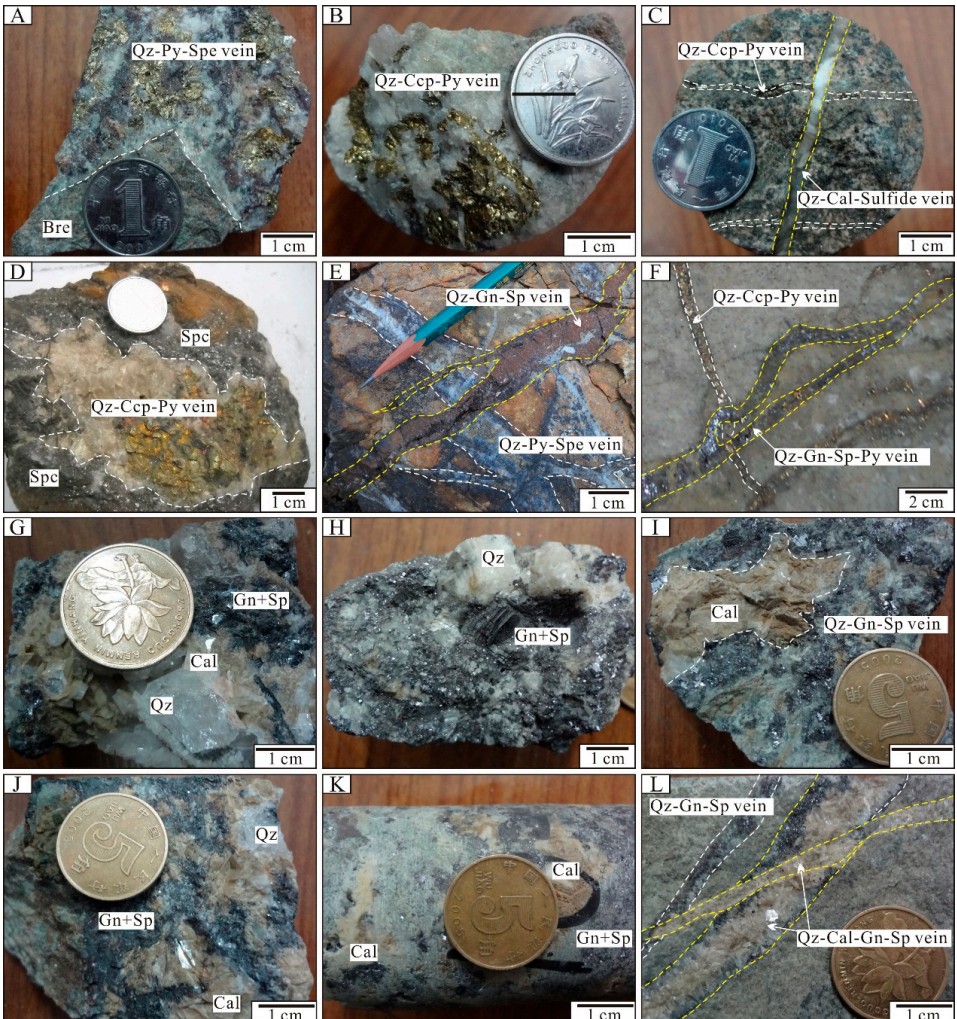

**Figure 5.** Photographs of ore from the Qibaoshan polymetallic ore field. (**A**) Quartz–pyrite–specularite vein from breccia-hosted ore in the Jinxiantou deposit; (**B**) Quartz–chalcopyrite–pyrite vein in the Jinxiantou deposit; (**C**) Quartz–chalcopyrite–pyrite vein cut by a later quartz–calcite–sulfide vein in the Jinxiantou deposit; (**D**) Quartz–pyrite–specularite vein cut by a later quartz–chalcopyrite–pyrite vein in the Changgou deposit; (**E**) Quartz–pyrite–specularite vein cut by a later quartz–sphalerite–galena vein in the Changgou deposit; (**F**) Quartz–chalcopyrite–pyrite vein cut by a later quartz–sphalerite–galena vein in the Changgou deposit; (**G**) Sphalerite–galena aggregate replaced by a later sulfide–poor–calcite aggregate in the Xingshanyu deposit; (**H**) Sphalerite–galena vein in the Xingshanyu deposit; (**I**) Sphalerite–galena vein cut by a later sulfide–poor–calcite vein in the Xingshanyu deposit; (**J**) Sphalerite–galena aggregate replaced by a later sulfide–poor–calcite aggregate in the Hongshigang deposit; (**K**) Sphalerite–galena vein cut by a later sulfide–poor–calcite vein in the Hongshigang deposit; and (**L**) Sphalerite–galena vein cut by a later sulfide–poor–calcite vein in the Hongshigang deposit. Abbreviations: Qz (quartz); Cal (calcite); Py (pyrite); Spe (specularite); Ccp (chalcopyrite); Sp (sphalerite); Gn (galena).

Based on the mineral assemblages and cross-cutting relationships, three stages of mineralization have been identified (Figure 5). Stage 1 mineralization was widespread and produced most of the Au. The characteristic minerals of this stage are milky-white quartz, pyrite, specularite, and gold. Stage 2 mineralization formed the quartz–pyrite–chalcopyrite veins and produced most of the Cu. Stage 3 mineralization produced quartz–calcite–pyrite veins, which have no economic value.

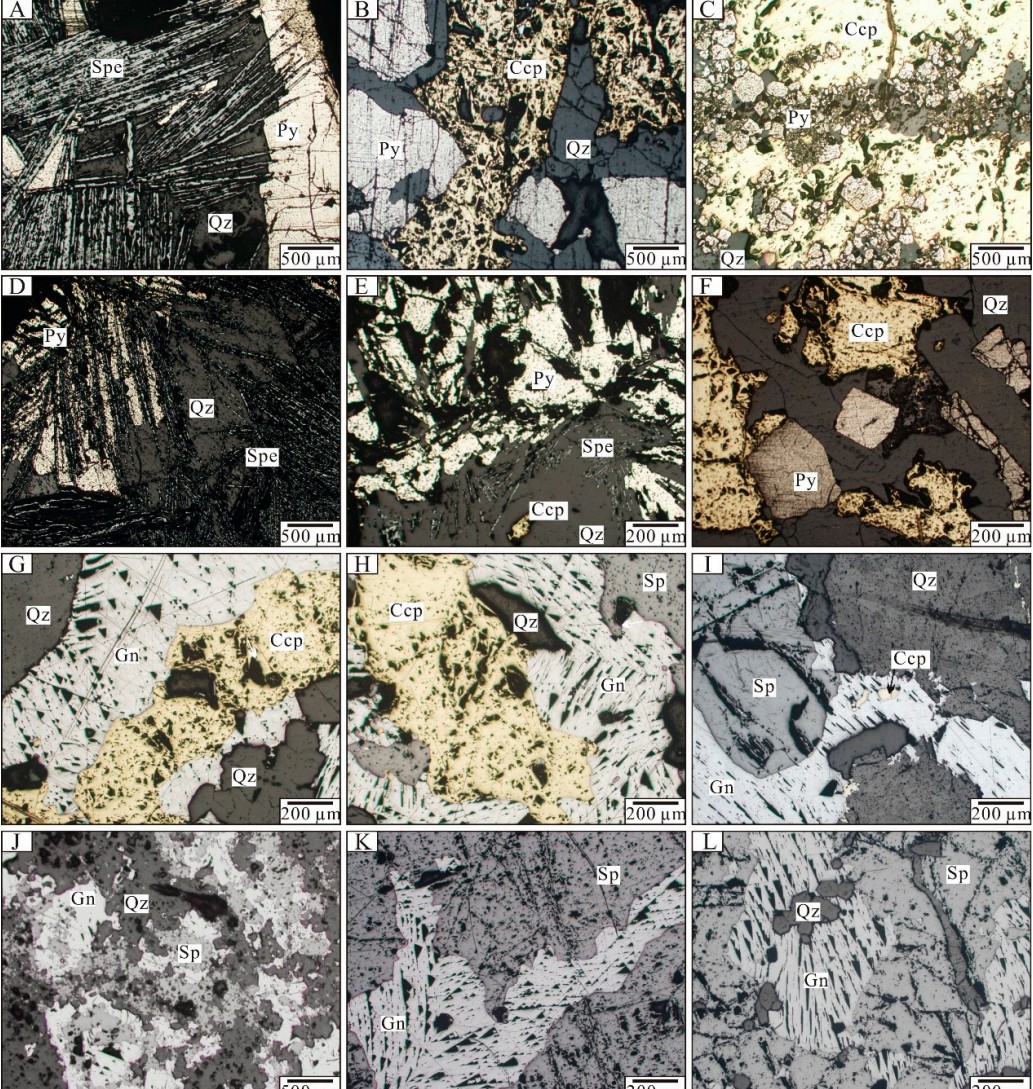

**Figure 6.** Photomicrographs showing the important mineral assemblages in the Qibaoshan polymetallic ore field. (**A**) Euhedral pyrite replaced by specularite in the Jinxiantou deposit; (**B,C**) Pyrite replaced by chalcopyrite in the Jinxiantou deposit; (**D,E**) Euhedral pyrite replaced by specularite in the Changgou deposit; (**F**) Pyrite replaced by chalcopyrite in the Changgou deposit; (**G,H**) Chalcopyrite replaced by sphalerite and galena aggregate in the Changgou deposit; (**I,J**) Sphalerite and galena aggregate in the Xingshanyu deposit; and (**K,L**) Sphalerite replaced by galena in the Hongshigang deposit. Abbreviations: Qz (quartz); Py (pyrite); Spe (specularite); Ccp (chalcopyrite); Sp (sphalerite); Gn (galena).

## 4.2. Changgou Deposit

The eastern section of the Changgou deposit is dominated by copper mineralization, while the western section is dominated by lead and zinc mineralization. A total of 10 ore bodies have been

discovered in the Changgou deposit, including eight Pb-Zn ore bodies and two Cu ore bodies. These ore bodies occur in the pyroxene diorites and andesites. They are mainly controlled by the E–W faults.

The C-1 ore body is the largest one in this deposit. It occurs within the pyroxene diorites and andesites. It is 908 m long and 2.2–4.1 m thick. It strikes NE–SW and dips 70°–80° SE (Figure 4B). Its ore grades are 1.39 wt. % Pb, 1.36 wt. % Zn, and 0.571 wt. % Cu.

The metal-bearing mineral assemblages of the ores include specularite, pyrite, chalcopyrite, galena, and sphalerite with minor digenite (Figures 5 and 6). The gangue minerals include quartz, calcite, epidote, kaolinite, and barite. The wall rock alteration is intense and consists of potassic alteration, silicification, sericitization, and carbonatization followed by epidotization.

The Changgou C-1 ore body contains numerous hydrothermal veins of various sizes. Based on the mineral assemblages and cross-cutting relationships, three stages of mineralization are recognized (Figure 5). The early mineralization stage formed the quartz–pyrite–specularite veins. The second stage of mineralization was widespread and produced most of the Cu. The characteristic minerals of this stage are milky-white quartz, pyrite, chalcopyrite, and minor digenite. The late-stage mineralization produced the quartz–galena–sphalerite veins, which are widespread and contain most of the Zn–Pb.

### 4.3. Hongshigang Deposit

A total of 12 Pb–Zn ore bodies have been discovered in the Hongshigang deposit. They occur in the pyroxene diorites, andesites, and diorite porphyries. They are mainly controlled by the NE–SW faults.

The H-1 ore body is the largest one in this deposit. It occurs in the pyroxene diorite and andesite. It is 908 m long and 2.2–4.1 m thick. It strikes NE–SW and dips 70° to ~80° SE (Figure 4C). Its ore grades are 0.59–2.49 wt. % Pb and 1.39 wt. % Zn.

The metal–bearing mineral assemblages of the ores include galena and sphalerite with minor pyrite and silver (Figures 5 and 6). The gangue minerals include ankerite, quartz, calcite, and kaolinite. The wall rocks of the ore bodies were subjected to beresitization, silicification, carbonation, chloritization, and sericitization.

Based on the ore mineralogy and cross-cutting relationships, the hypogene fissure-filling mineralization can be divided into two paragenetic stages (Figure 5). Stage 1 mineralization consists of quartz–galena–sphalerite veins (Figure 3A), which are widespread and contain most of the Pb–Zn. Stage 2 mineralization consists of quartz–calcite–sulfide–poor veins, which cut across the early galena–sphalerite–pyrite veins.

### 4.4. Xingshanyu Deposit

A total of 15 Pb–Zn ore bodies have been discovered in the Xingshanyu deposit. Most of these ore bodies are exposed on the surface, but some of them are concealed. They occur in the andesites, diorite porphyries, and in the Qingshan Group, and they are mainly controlled by the NE–SW and NW-SE faults.

The X-1 ore body is the largest one in this deposit. It is located in the Qingshan Group. It is 1075 m long and 1.96–5.45 m thick. It strikes NE–SW and dips 49°–82° (Figure 4A). Its ore grades are 0.39–1.05 wt. % Pb and 0.75 wt. % Zn.

The assemblage of sulfide minerals includes galena, sphalerite, minor silver, and pyrite (Figures 5 and 6). The gangue minerals include quartz, calcite, dolomite, and kaolinite. Wall rock alteration is intense and consists of potassic alteration, beresitization, kaolinization, silicification, and carbonation. Various types of hydrothermal alteration are spatially superimposed on both sides of the vein.

Based on the ore mineralogy and cross-cutting relationships, two paragenetic stages occurred, which are similar to the stages of the Hongshigang deposit (Figure 5). The early mineralizationconsists of quartz–galena–sphalerite veins are widespread and contain most of the

Pb–Zn. The late mineralization stage consists of quartz–calcite–sulfide–poor veins, which cut across the early quartz–galena–sphalerite veins.

## 5. Samples and Analytical Methods

The samples in this study were mainly collected from the drill holes and underground mining tunnels in the Jinxiantou, Changgou, Hongshigang, and Xingshanyu deposits.

### 5.1. Fluid Inclusions

Sixty-five doubly-polished thin sections (approximately 0.20–0.30 mm thick) were prepared from quartz samples from the different mineralization stages of the four deposits. FI petrography involved careful observation of the shapes, the spatial distribution characteristics, the genetic and compositional types, and the vapor/liquid ratios. Thirty-two samples with abundant representative FIs were selected for microthermometric measurements. The FI microthermometric analyses were performed using a Linkam THMS600 heating–freezing stage (Linkam Scientific, Surrey, UK) with a temperature range of −196 °C to 600 °C. Calibration of the stage was completed using the following standards: pure water inclusions (0 °C), pure $CO_2$ inclusions (−56.6 °C), and potassium bichromate (398 °C). This yielded an accuracy of ±0.2 °C during freezing and heating. The fluid salinities (NaCl equivalent) of the inclusions were calculated using the final melting temperature of the ice [25]. These FI studies were conducted at the Geological Fluid Laboratory, College of Earth Science, Jilin University, China.

Quartz, galena, and chalcopyrite grains from the different mineralization stages were handpicked from the 40–60 mesh size fraction under a binocular microscope (purity > 99%). Hydrogen, oxygen and sulfur isotopic analyses were performed using a MAT-253 mass spectrometer at the Analytical Laboratory of Beijing Research Institute of Uranium Geology, China National Nuclear Corporation (CNNC).

### 5.2. H–O–S Isotopes

The oxygen isotopic analyses were performed on 10 to 20-mg samples of quartz using the bromine pentafluoride method of Clayton and Mayeda [26], and were converted to $CO_2$ on a platinum coated carbon rod. The $\delta^{18}O_{FI}$ values of the FI water from the quartz samples were calculated using the formula ($\delta^{18}O_{FI} = \delta^{18}O_{qz} − 1000\ln\alpha_{qz\text{-water}}$, $1000\ln\alpha_{qz\text{-water}} = 3.38 \times 10^6 T^{-2} − 3.4$ [27]). In this formula, T is the mineralization temperature, which is approximately equal to the average homogenization temperature. The hydrogen isotopic composition of the FI water was determined by decrepitation of the FIs within the quartz samples at 600 °C for 30 min. The water was reduced to $H_2$ by passing it over a uranium metal-bearing tube. The $H_2$ was collected by sample collection tube and then transferred to the mass spectrometer for analysis. The sulfides were reacted with $Cu_2O$ until they transformed into pure $SO_2$. The same sample was analyzed three times, and the analytical precisions were ±0.2‰ for $\delta^{18}O$, ±2‰ for $\delta D$ and ±2‰ for $\delta^{34}S$. Then, we tested the standard sample Vienna Standard Mean Ocean Water (VSMOW) and Vienna-Canyon Diablo Troilite (VCDT) employing the same procedure to calibrate the sample measurement. All the $\delta^{18}O$ and $\delta D$ values reported in this paper are normalized versus VSMOW; the $\delta^{34}S$ values are normalized versus VCDT scales.

## 6. Results

### 6.1. Fluid Inclusion Petrography

The criteria established by Roedder [28] and Hollister and Burruss [29] were used to distinguish different generations of FIs within the hydrothermal quartz. The primary inclusions were isolated or occurred in random groups, while the secondary inclusions filled the microcracks. Each cluster or group of inclusions along the growth bands was considered to represent a separate fluid inclusion

assemblage (FIA). Populations of different types of FIs were identified based on their room temperature phase relationships and their phase transitions during heating and cooling.

Three types of FIs were identified using the nomenclature of Shepherd et al. [30]: liquid-rich aqueous (LV type) inclusions, vapor-rich aqueous (V type) inclusions, and halite-bearing (S type) inclusions (Figure 7).

The lack of liquid $CO_2$ or clathrate formation during freezing indicates that none of the inclusions contained significant quantities of $CO_2$. The content of $CO_2$ was less in the gas phase composition (16.7 µL/g), as reported by Wang et al. [1].

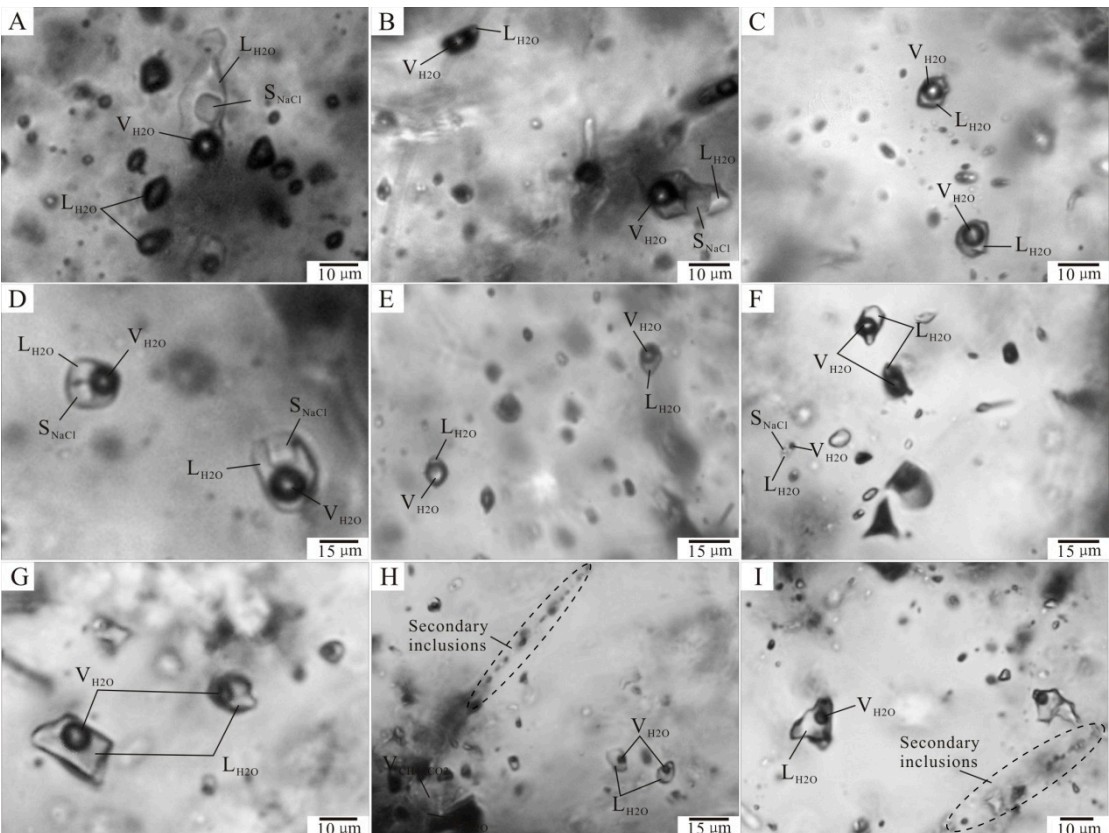

**Figure 7.** Photomicrographs of FIs in the Qibaoshn polymetallic ore field. (**A–C**) Liquid-rich aqueous (LV) type, halite-bearing (S) type, and vapor-rich aqueous (V) type fluid inclusions (FIs) in quartz from the Jinxiantou deposit; (**D–F**) S type, LV type, and V type FIs in quartz from the Changgou deposit; (**G**) LV type FIs in quartz from the Xingshanyu deposit; and (**H–I**) LV type FIs in quartz from the Hongshigang deposit. Abbreviations: L (liquid phase), V (vapor phase), S (halite crystal).

LV-type FIs consisted of a vapor bubble and a liquid phase at room temperature. These FIs were typically rectangular or elliptical and between 6–12 µm in size. The vapor bubbles accounted for 15–30% of the total inclusion volume. These inclusions were present in all the stages of mineralization and commonly occurred as planar arrays restricted to the interiors of the quartz crystals. However, some LV-type inclusions filled the microfractures in the quartz, indicating a secondary origin.

V-type FIs consisted of a single vapor phase, or a liquid with a vapor bubble, which accounted for 70–95% of the inclusion volume. This type of FI was typically elliptical to sub-rounded, and was between 8–10 µm.

S-type FIs contained three phases at room temperature: a vapor bubble, liquid water, and a halite cube. These inclusions were always <20 µm and occurred in isolation or as discrete clusters, implying a primary origin.

### 6.2. Fluid Inclusion Microthermometry

Fluid inclusion assemblages were chosen for microthermometric analysis. Data for all of the stages from the four deposits are listed in Table 1. Histograms of the homogenization temperatures ($T_h$) and salinities of the different types of FIs in the various quartz stages of the four deposits are presented in Figure 8.

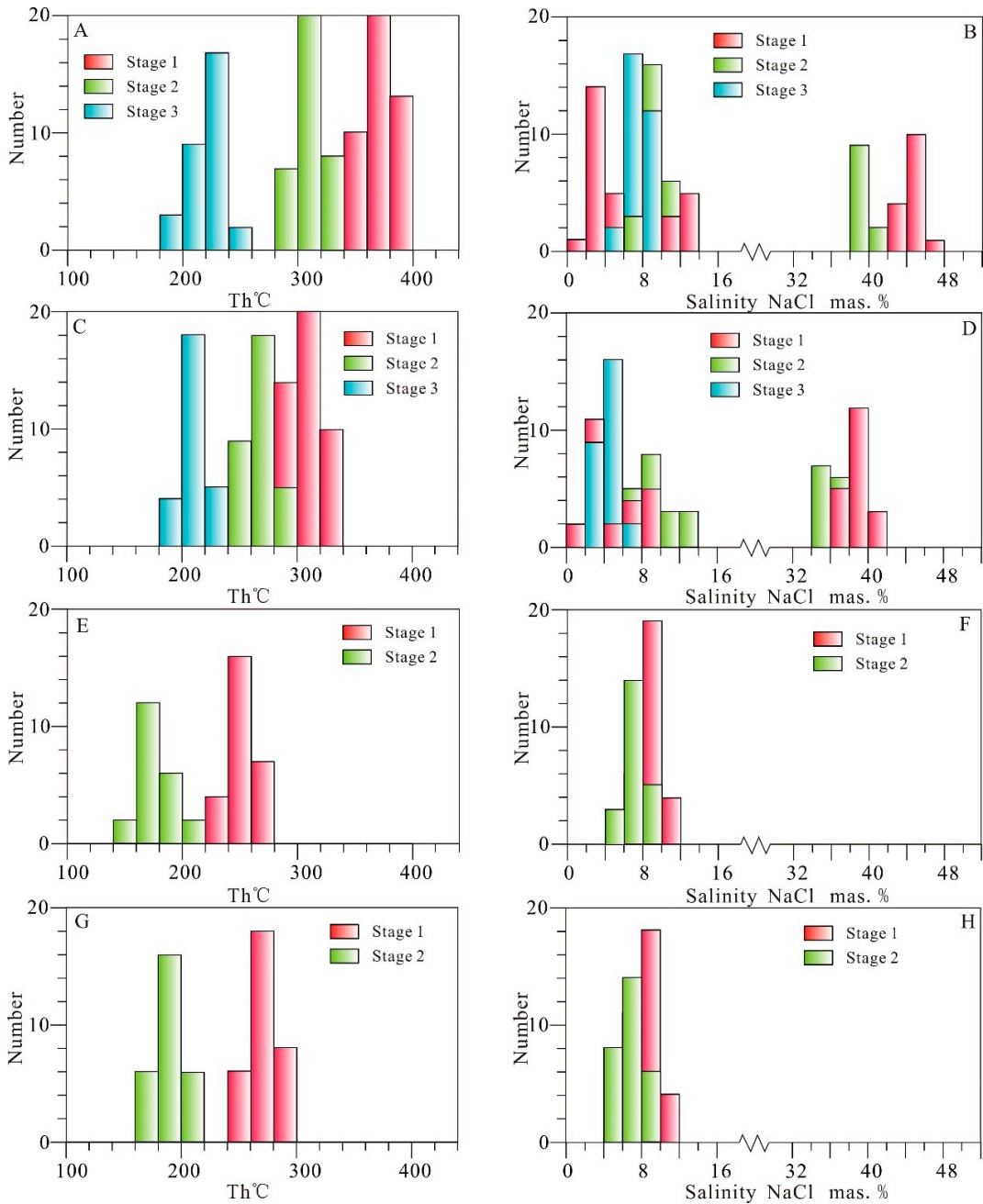

**Figure 8.** Histograms of microthermometric data for the FIs in the Qibaoshn polymetallic ore field. (**A**,**B**) Jinxiantou deposit; (**C**,**D**) Changgou deposit; (**E**,**F**) Hongshigang deposit; (**G**,**H**) Xingshanyu deposit.

**Table 1.** Microthermometric data for fluid inclusions (FIs) from hydrothermal quartz from the Qibaoshan polymetallic ore field.

| Deposit | Stage | Sample (Quartz) | FIA | FI Type | Number of Inclusion Measured | $T_e$ (°C) | $T_m$ (ice) (°C) | $T_m$ (NaCl) (°C) | Salinity (mas. % NaCl) | $T_h$ (°C) |
|---|---|---|---|---|---|---|---|---|---|---|
| Jinxiantou | I | JXT-1 | V and S | V | 6 | | −2.8 to −1.2 | | 2.1–4.6 | 366–395 |
| | | | | S | 5 | | | 361–389 | 43.4-46.3 | 365–389 |
| | | JXT-3 | V | V | 9 | | −2.7 to −1.1 | | 1.9–4.5 | 372–397 |
| | | JXT-4 | LV | LV | 8 | −26.5 to −28.1 | −10.1 to −7.6 | | 11.2–14.0 | 351–375 |
| | | JXT-6 | V and S | V | 5 | | −2.3 to −1.3 | | 2.2–3.9 | 375–386 |
| | | JXT-7 | | S | 10 | | | 355–385 | 42.9–45.8 | 362–385 |
| | II | JXT-11 | LV | LV | 13 | −24.7 to −27.8 | −7.8 to −4.8 | | 7.6–11.5 | 285–319 |
| | | JXT-12 | LV | LV | 11 | −25.3 to −26.4 | −8.2 to −5.7 | | 8.8–12.0 | 298–326 |
| | | JXT-15 | S | S | 5 | | | 300–313 | 38.2–39.2 | 302–315 |
| | | JXT-16 | S | S | 6 | | | 319–330 | 39.7–40.6 | 319–332 |
| | III | JXT-18 | LV | LV | 10 | −23.7 to −24.5 | −6.0 to −3.9 | | 6.3–9.2 | 191–226 |
| | | JXT-19 | LV | LV | 12 | −23.9 to −25.8 | −6.1 to −4.1 | | 6.6–9.3 | 202–242 |
| | | JXT-20 | LV | LV | 9 | −24.2 to −26.1 | −5.6 to −3.6 | | 5.9–8.7 | 196–227 |
| Changgou | I | CG-1 | LV | LV | 9 | −25.7 to −27.8 | −5.9 to −4.1 | | 6.6–9.1 | 286–326 |
| | | CG-2 | V and S | V | 6 | | −1.4 to −1.0 | | 1.7–2.4 | 309–315 |
| | | | | S | 3 | | | 280–310 | 36.7–38.9 | 302–312 |
| | | CG-4 | S | S | 8 | | | 292–325 | 37.6–40.2 | 295–328 |
| | | CG-6 | V and S | V | 9 | | −2.5 to −1.2 | | 2.1–4.2 | 315–325 |
| | | | | S | 11 | | | 295–323 | 37.8–40.0 | 299–322 |
| | II | CG-10 | LV and S | LV | 6 | −24.7 to −26.8 | −11.2 to −8.5 | | 12.3–15.2 | 261–285 |
| | | | | S | 4 | | | 249–270 | 34.6–36.0 | 259–272 |
| | | CG-13 | S | S | 9 | | | 261–293 | 35.4–37.6 | 265–293 |
| | | CG-15 | LV | LV | 13 | −25.0 to −27.3 | −7.6 to −5.8 | | 8.9–11.2 | 245–281 |
| | III | CG-16 | LV | LV | 11 | −23.7 to −25.6 | −4.9 to −3.4 | | 5.6–7.7 | 189–226 |
| | | CG-19 | LV | LV | 16 | −23.2 to −26.1 | −5.3 to −4.4 | | 6.6–8.3 | 192–235 |
| Hongshigang | I | HSG-1 | LV | LV | 10 | −29.1 to −31.6 | −6.2 to −4.6 | | 7.3–9.5 | 236–252 |
| | | HSG-2 | LV | LV | 8 | −28.9 to −32.5 | −6.9 to −5.5 | | 8.5–10.4 | 240–265 |
| | | HSG-6 | LV | LV | 11 | −31.5 to −34.5 | −6.5 to −4.5 | | 7.2–9.9 | 247–273 |
| | II | HSG-7 | LV | LV | 10 | −30.9 to −32.6 | −4.9 to −3.6 | | 5.9–7.7 | 155–183 |
| | | HSG-8 | LV | LV | 12 | −28.7 to −31.8 | −5.4 to −3.8 | | 6.2–8.4 | 169–202 |
| Xingshanyu | I | XSY-1 | LV | LV | 10 | −30.3 to −34.8 | −6.3 to −4.8 | | 7.6–9.6 | 246–269 |
| | | XSY-2 | LV | LV | 15 | −29.2 to −32.3 | −7.0 to −5.3 | | 8.3–10.5 | 258–289 |
| | | XSY-5 | LV | LV | 8 | −28.9 to −31.9 | −6.8 to −4.8 | | 7.6–10.2 | 252–281 |
| | II | XSY-6 | LV | LV | 7 | −29.2 to −31.6 | −4.9 to −3.6 | | 5.9–7.7 | 165–186 |
| | | XSY-8 | LV | LV | 13 | −28.6 to −30.7 | −5.5 to −3.9 | | 6.3–8.5 | 175–209 |
| | | XSY-11 | LV | LV | 8 | −30.2 to −33.4 | −5.2 to −3.4 | | 5.6–8.1 | 172–198 |

Abbreviations: FIA (fluid inclusion assemblage); LV, V, and S (fluid inclusion types); $T_e$ (eutectic temperatures); $T_m$ (ice) (final ice melting temperature); $T_m$ (NaCl) (final halite crystals dissolving temperatures in the inclusions); $T_h$ (homogenization temperature).

6.2.1. Jinxiantou Deposit

The Stage 1 quartz veins contained abundant LV, V, and S-type FIs. The eutectic temperatures ($T_e$) of the LV-type FIs ranged from −28.1 to −26.5 °C. The final ice-melting temperatures of the LV-type FIs ranged from −10.1 to −7.6 °C, corresponding to salinities of 11.2–14.0 mas. % NaCl equivalent. Total homogenization of the LV-type FIs to the liquid phase occurred at temperatures of 351–375 °C. The final ice-melting temperatures of the V-type FIs ranged from −2.8 to −1.1 °C, corresponding to salinities of 1.9–4.5 mas. % NaCl equivalent. Total homogenization of the V-type FIs to the vapor phase occurred at temperatures of 372–397 °C.

Upon heating, there are three mechanisms by which the S-type FIs can become homogenized: (1) The vapor phase disappeared first; then, the daughter minerals dissolved, and the FIs homogenized to a single liquid phase. (2) The daughter minerals dissolved followed by the disappearance of the vapor phase, and the FIs homogenized to a single liquid phase. (3) The vapor phase and daughter minerals disappeared simultaneously. The third homogenization mechanism is considered to be the most important, because it includes the homogenization of the FIs to a single liquid phase, which typically occurs at temperatures of 362–389 °C. The halite crystals in the inclusions dissolved at 255–389 °C, corresponding to salinities of 32.9–46.3 mas. % NaCl equivalent.

The Stage 2 quartz veins contained LV and S-type FIs. The $T_e$ values of the LV-type FIs ranged from −27.8 to −24.7 °C. The final ice-melting temperatures of the LV type FIs ranged from −8.2 to −4.8 °C, corresponding to salinities of 7.6–12.0 mas. % NaCl equivalent. Total homogenization of the LV-type FIs to the liquid phase occurred at temperatures of 285–326 °C. The halite crystals in the S-type FIs dissolved at 300–330 °C, corresponding to salinities of 38.2–40.6 mas. % NaCl equivalent. These inclusions homogenized to the liquid phase at temperatures of 302–332 °C.

The Stage 3 quartz veins contained only LV-type FIs, which had the lowest $T_h$ values and salinities of the FIs from the Jinxiantou deposit. The $T_e$ values of the LV type FIs ranged from −26.1 to −23.7 °C. Homogenization of FIs to the liquid phase occurred at 191–242°C, and the final ice melting temperatures ranged from −6.1 to −3.6 °C, corresponding to salinities of 5.9–9.3 mas. % NaCl equivalent.

6.2.2. Changgou Deposit

The Stage 1 quartz veins contained abundant LV, V, and S-type FIs. The $T_e$ values of the LV-type FIs ranged from −27.8 to −25.7 °C. The final ice-melting temperatures of the LV-type FIs ranged from −5.9 to −4.1 °C, corresponding to salinities of 6.6–9.1 mas. % NaCl equivalent. Total homogenization of the LV-type FIs to the liquid phase occurred at temperatures of 283–326 °C. The final ice-melting temperatures of the V-type FIs ranged from −2.5 to −1.0 °C, corresponding to salinities of 1.7–4.2 mas. % NaCl equivalent. Total homogenization of the V-type FIs to the vapor phase occurred at temperatures of 299–322 °C. Upon heating, the S-type FIs also experienced three modes of homogenization similar to those in the Jinxiantou deposit. The FIs homogenized to a single liquid phase at temperatures of 295–328 °C. The halite crystals within the inclusions dissolved at temperatures of 280–325 °C, corresponding to salinities of 36.7–40.2 mas. % NaCl equivalent.

The Stage 2 quartz veins contained LV-type and S-type FIs. The $T_e$ values of the LV-type FIs ranged from −27.3 to −25.7 °C. The final ice-melting temperatures of the LV-type FIs ranged from −11.2 to −5.8 °C, corresponding to salinities of 8.9–15.2 mas. % NaCl equivalent. Total homogenization of the LV-type FIs to the liquid phase occurred at temperatures of 245–285 °C. The halite crystals in the S-type FIs dissolved at 249–293 °C, corresponding to salinities of 34.6–37.6 mas. % NaCl equivalent. These inclusions homogenized to the liquid phase at 259–293 °C.

The Stage 3 quartz veins contained only LV-type FIs, which had the lowest recorded $T_h$ values and salinities of the Changgou deposit. The $T_e$ values of the LV type FIs ranged from −26.1 to −23.2 °C. Homogenization to the liquid phase occurred at 189–235 °C, and the final ice melting temperatures ranged from −5.3 to −3.4 °C, corresponding to salinities of 5.6–8.3 mas. % NaCl equivalent.

### 6.2.3. Hongshigang Deposit

The Stage 1 quartz veins contain only LV-type FIs. The $T_e$ values of the LV type FIs ranged from −34.5 to −28.9 °C. Their final ice-melting temperatures ranged from −6.9 to −4.5 °C. The estimated salinities range from 7.2 to 10.4 mas. % NaCl equivalent. The FIs homogenized to the liquid phase at temperatures of 236 to 273 °C.

The Stage 2 quartz veins also contained LV-type FIs. The $T_e$ values of the LV-type FIs ranged from −32.6 to −28.7 °C. The final ice-melting temperatures ranged from −5.4 to −3.6 °C. The estimated salinities ranged from 5.9 to 8.4 mas. % NaCl equivalent. The FIs homogenized to the liquid phase at temperatures of 155 to 202 °C.

### 6.2.4. Xingshanyu Deposit

The Stage 1 quartz veins contained only LV-type FIs. The $T_e$ values of the LV type FIs ranged from −34.8 to −28.9 °C. The final ice-melting temperatures of the LV type FIs ranged from −7.0 to −4.8 °C, corresponding to salinities of 7.6–10.5 mas. % NaCl equivalent. Total homogenization of the LV-type FIs to the liquid phase occurred at temperatures of 246–289 °C.

The Stage 2 quartz veins also contained LV-type FIs. The $T_e$ values of the LV-type FIs ranged from −33.4 to −28.6 °C. The final ice-melting temperatures of the LV type FIs ranged from −5.5 to −3.4 °C, corresponding to salinities of 5.6–8.5 mas. % NaCl equivalent. Total homogenization of the LV-type FIs to the liquid phase occurred at temperatures of 165 to 209 °C.

### 6.3. Oxygen and Hydrogen Isotopes

The stable isotope (H, O) data for the water extracted from the FIs was used to determine the source of the ore-forming fluids. The H and O isotopic data obtained in this study is unique because this is the first time that these isotopes have been measured for the fluid phase of FIs in hydrothermal quartz from the four deposits of the Qibaoshan polymetallic ore field.

Fluid inclusion water from the quartz samples from the Jinxiantou, Changgou, Hongshigang, and Xingshanyu deposits had $\delta D_{FI}$ values of −106.3‰ to −88.6‰, −115.6‰ to −101.2‰, −109.8‰ to −100.2‰, and −108.3‰ to −100.6‰; and $\delta^{18}O_{qz}$ values of 11.1‰ to 12.3‰, 12.2‰ to 13.4‰, 10.5‰ to 11.8‰, and 10.2‰ to 12.1‰, respectively (Table 2). The calculated $\delta^{18}O_{FI}$ values range from 1.8‰ to 7.0‰, −1.1‰ to 6.7‰, −3.5‰ to 3.2‰, and −2.0‰ to 4.0‰, for the Jinxiantou, Changgou, Hongshigang, and Xingshanyu deposits, respectively.

**Table 2.** Oxygen and hydrogen isotopic data for quartz and fluid inclusion waters associated with various stages of hydrothermal quartz from the Qibaoshan polymetallic ore field.

| Deposit | Sample | Stage | $\delta^{18}O_{qz}$ (‰ V-SMOW) | $\delta D_{FI}$ (‰ V-SMOW) | $T_h$ (°C) | $\delta^{18}O_{FI}$ (‰ V-SMOW) |
|---|---|---|---|---|---|---|
| | JT-13 | 1 | 11.5 | −88.6 | 371 ± 15 | 6.8 ± 0.4 |
| | JT-15 | 1 | 11.9 | −91.8 | 371 ± 15 | 7.2 ± 0.4 |
| Jinxiantou | JT-16 | 2 | 11.8 | −102.5 | 312 ± 15 | 5.3 ± 0.5 |
| | JT-17 | 2 | 11.1 | −103.9 | 312 ± 15 | 4.6 ± 0.5 |
| | JT-19 | 3 | 12.2 | −104.6 | 223 ± 16 | 1.8 ± 0.8 |
| | JT-22 | 3 | 12.3 | −106.3 | 223 ± 16 | 1.9 ± 0.8 |
| | CG-12 | 1 | 12.9 | −101.9 | 307 ± 17 | 6.3 ± 0.6 |
| | CG-13 | 1 | 13.1 | −101.2 | 307 ± 17 | 6.5 ± 0.6 |
| | CG-15 | 1 | 12.6 | −104.5 | 307 ± 17 | 6.0 ± 0.6 |
| | CG-17 | 1 | 12.4 | −104.1 | 307 ± 17 | 5.8 ± 0.6 |
| | CG-18 | 2 | 12.9 | −105.2 | 268 ± 14 | 4.8 ± 0.6 |
| | CG-19 | 2 | 12.5 | −105.3 | 268 ± 14 | 4.4 ± 0.6 |
| Changgou | CG-20 | 2 | 12.2 | −106.2 | 268 ± 14 | 4.1 ± 0.6 |
| | CG-22 | 2 | 12.3 | −103.8 | 268 ± 14 | 4.2 ± 0.6 |
| | CG-23 | 3 | 12.7 | −107.1 | 212 ± 15 | 1.7 ± 0.8 |
| | CG-25 | 3 | 13.4 | −104 | 212 ± 15 | 2.4 ± 0.8 |
| | CG-26 | 3 | 12.2 | −115.6 | 212 ± 15 | 1.2 ± 0.8 |
| | CG-27 | 3 | 12.8 | −110.2 | 212 ± 15 | 1.8 ± 0.8 |
| | HS-11 | 1 | 10.5 | −109.8 | 250 ± 15 | 1.6 ± 0.7 |
| | HS-13 | 1 | 10.8 | −106.6 | 250 ± 15 | 1.9 ± 0.7 |
| Hongshigang | HS-14 | 1 | 11.7 | −102.4 | 250 ± 15 | 2.8 ± 0.7 |
| | HS-15 | 1 | 11.4 | −100.2 | 250 ± 15 | 2.5 ± 0.7 |

| | HS-18 | 2 | 10.9 | −101.9 | 177 ± 16 | −2.4 ± 0.6 |
|---|---|---|---|---|---|---|
| | HS-19 | 2 | 11.8 | −105.2 | 177 ± 16 | −1.5 ± 0.6 |
| | HS-21 | 2 | 11.7 | −103.6 | 177 ± 16 | −1.6 ± 0.6 |
| | HS-23 | 2 | 11.1 | −107.1 | 177 ± 16 | −2.2 ± 0.6 |
| | XS-11 | 1 | 11.3 | −100.6 | 270 ± 15 | 3.2 ± 0.6 |
| | XS-12 | 1 | 10.9 | −102.1 | 270 ± 15 | 2.8 ± 0.6 |
| | XS-14 | 1 | 11.6 | −101.4 | 270 ± 15 | 3.5 ± 0.6 |
| Xingshanyu | XS-18 | 1 | 12.1 | −103.2 | 270 ± 15 | 4.0 ± 0.6 |
| | XS-20 | 2 | 11.4 | −104.9 | 191 ± 15 | −0.9 ± 0.4 |
| | XS-21 | 2 | 11.9 | −107.5 | 191 ± 15 | −0.4 ± 0.4 |
| | XS-22 | 2 | 10.2 | −108.3 | 191 ± 15 | −2.1 ± 0.4 |
| | XS-25 | 2 | 11.1 | −105.2 | 191 ± 15 | −1.2 ± 0.4 |

Abbreviations: $T_h$ = mean homogenization temperature at 1<sigma>.

### 6.4. Sulfur Isotopes

We obtained new S isotopic data for the Qibaoshan polymetallic ore field. The data is reported in Table 3. The $\delta^{34}S$ values of the chalcopyrite in the Jinxiantou and Changgou deposits ranged from 2.4‰ to 4.3‰ and from 2.9‰ to 4.2‰, respectively. The $\delta^{34}S$ values of the galena in the Hongshigang and Xingshanyu deposits ranged from 0.6‰ to 3.0‰ and 0.8‰ to 3.2‰, for the Jinxiantou, Changgou, Hongshigang, and Xingshanyu deposits, respectively.

## 7. Discussion.

### 7.1. Sources of Ore Constituents

Sulfur isotopes are important tools for determining the source (s) of ore-forming materials in deposits [31–33]. In this study, the $\delta^{34}S$ values of the sulfides in the Qibaoshan polymetallic ore field had a limited range of 0.5‰ to 4.3‰ (n = 18). However, there are differences in the $\delta^{34}S$ values of the different deposits. The data for these deposits is close to the $\delta^{34}S$ values (−3‰ to 3‰) reported for the granitic rocks [33,34], suggesting that a magma source may have contributed to the mineralization (Figure 9). The higher $\delta^{34}S$ values (>3‰) of some of the sulfides are likely due to the influence of the wall rocks, which have high $\delta^{34}S$ values in this region, i.e., 6.1–10.1‰ for the Linglong granite [35] and 2.7–10.0‰ for the Guojialing granite [36].

Both the quartz–pyrite–chalcopyrite assemblage observed in the Jinxiantou and Changgou deposits and the quartz–galena–sphalerite–pyrite mineral assemblage observed in the Hongshigang and Xingshanyu deposits lacked an oxidized phase. A lack of sulfate minerals indicates that sulfur was present in the hydrothermal fluids mainly as reduced sulfur ($H_2S$). The $\delta^{34}S_{VCDT}$ values of the $H_2S$ ($\delta^{34}S_{H2S}$) in equilibrium with the sulfides were estimated by evaluating the $\delta^{34}S_{VCDT}$ values of the sulfides ($\delta^{34}S_{sulfide}$) and the mineralization temperature of the hydrothermal fluids using the formula $A \times 10^6/T^2 = \delta^{34}S_{sulfide} - \delta^{34}S_{H2S}$ [37]. In the formula, T is the mineralization temperature, and A is the equilibrium isotopic fractionation factor ($A_{chalcopyrite}$ = −0.05, $A_{galena}$ = −0.63). During homogeneous trapping, the homogenization temperature of a fluid inclusion is the minimum trapping temperature, and the true formation temperature of the fluid inclusion may be higher. Since the range of homogenization temperatures of the FIs in the four deposits was limited, we use the mean value and perform bounding calculations using standard deviation to limit the mineralization temperatures. The calculated $\delta^{34}S_{H2S}$ values in the hydrothermal fluids are 2.8‰ to 5.3‰, 2.6‰ to 5.3‰, 3.1‰ to 4.4‰, and 2.5‰ to 4.4‰, respectively (Table 3).

**Table 3.** Sulfur isotopic data for ore minerals from the Qibaoshan polymetallic ore field.

| Deposit | Sample | Stage | Mineral | $\delta^{34}S_{sulfide}$ (‰) | $\delta^{34}S_{H2S(mean)}$ (‰) | $T_{h(mean)}$ (°C) | Sample Location |
|---|---|---|---|---|---|---|---|
| Jinxiantou | JT-1-2 | 2 | Chalcopyrite | 4.3 | 4.4 ± 0.01 | 312 ± 15 | ZK64, −42 m |
| Jinxiantou | JT-4-2 | 2 | Chalcopyrite | 3.4 | 3.5 ± 0.01 | 312 ± 15 | ZK141, −150 m |
| Jinxiantou | JT-7-2 | 2 | Chalcopyrite | 2.4 | 2.5 ± 0.01 | 312 ± 15 | ZK143, −545 m |
| Jinxiantou | JT-8 | 2 | Chalcopyrite | 2.9 | 3.0 ± 0.01 | 312 ± 15 | ZK143, −451 m |
| Changgou | CG-1-1 | 2 | Chalcopyrite | 4.2 | 4.4 ± 0.01 | 268 ± 14 | ZK706-1, 116 m |
| Changgou | CG-1-2 | 2 | Chalcopyrite | 4.2 | 4.4 ± 0.01 | 268 ± 14 | ZK706-1, 122 m |

| Changgou | CG-10-1 | 2 | Chalcopyrite | 3.3 | 3.5 ± 0.01 | 268 ± 14 | ZK706-2, −148 m |
|---|---|---|---|---|---|---|---|
| Changgou | CG-10-2 | 2 | Chalcopyrite | 3.0 | 3.2 ± 0.01 | 268 ± 14 | ZK706-2, −153 m |
| Changgou | CG-9-1 | 2 | Chalcopyrite | 3.8 | 4.0 ± 0.01 | 268 ± 14 | ZK706, 135 m |
| Changgou | CG-9-2 | 2 | Chalcopyrite | 2.9 | 3.1 ± 0.01 | 268 ± 14 | ZK706, 146 m |
| Hongshigang | HS-1-1 | 1 | Galena | 0.6 | 2.9 ± 0.07 | 250 ± 15 | ZK3001, 5 m |
| Hongshigang | HS-1-2 | 1 | Galena | 0.5 | 2.8 ± 0.07 | 250 ± 15 | ZK3001, 15 m |
| Hongshigang | HS-2-1 | 1 | Galena | 3.0 | 5.3 ± 0.07 | 250 ± 15 | ZK0001, −38 m |
| Hongshigang | HS-2-2 | 1 | Galena | 2.7 | 5.0 ± 0.07 | 250 ± 15 | ZK0001, −40 m |
| Hongshigang | HS-3-1 | 1 | Galena | 1.8 | 4.1 ± 0.07 | 250 ± 15 | ZK0002, −225 m |
| Hongshigang | HS-3-2 | 1 | Galena | 2.5 | 4.8 ± 0.07 | 250 ± 15 | ZK0002, −236 m |
| Xingshanyu | XS-1 | 1 | Galena | 0.8 | 2.9 ± 0.12 | 270 ± 15 | ZK5-1, 52 m |
| Xingshanyu | XS-2 | 1 | Galena | 1.2 | 3.3 ± 0.12 | 270 ± 15 | ZK5-1, 48 m |
| Xingchanyu | XS-3 | 1 | Galena | 1.5 | 3.6 ± 0.12 | 270 ± 15 | ZK5-2, 145 m |
| Xingshanyu | XS-4 | 1 | Galena | 2.8 | 4.9 ± 0.12 | 270 ± 15 | ZK5-2, 153 m |
| Xingchanyu | XS-5 | 1 | Galena | 3.2 | 5.3 ± 0.12 | 270 ± 15 | ZK5, 107 m |
| Xingshanyu | XS-6 | 1 | Galena | 2.4 | 4.5 ± 0.12 | 270 ± 15 | ZK5, 113 m |

Abbreviations: $A \times 10^6/T^2 = \delta^{34}S_{sulfide} - \delta^{34}S_{H2S}$ [37], $T$ = mineralization temperature = mean homogenization temperature, $A_{chalcopyrite} = -0.05$, $A_{galena} = -0.63$.

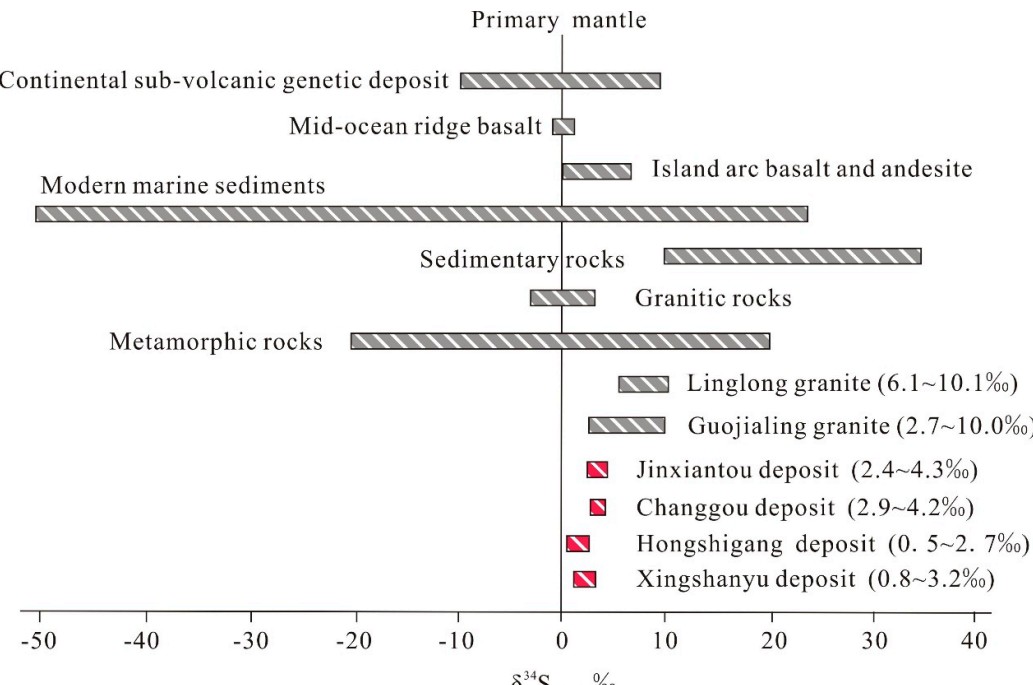

**Figure 9.** $\delta^{34}S$ values of quartz from the Qibaoshan polymetallic ore field compared with the values of important geological reservoirs. The data for Linglong granite and Guojialing granite are from Yang [35] and Li et al. [36], respectively. Other data are from Hoefs [33].

The $\delta^{34}S_{sulfide}$ values that we measured for the sulfides and the $\delta^{34}S_{H2S}$ values we calculated for the H₂S suggest a mixing source, with the sulfur produced primarily by a magmatic source with a lesser contribution from wall rock contamination, such as Linglong granite, Guojialing granite, and other wall rocks with high positive $\delta^{34}S$ values [38].

### 7.2. Fluid Compositions and Pressure–Temperature (P–T) Conditions of Fluid Trapping

In the Jinxiantou and Changgou deposits, the $T_e$ values of the FIs ranged from −23.2 to −28.1 °C, which suggests that the fluid is mainly composed of Na-chloride solutions with low concentrations of other cations (K, Mg) [27]. Fluid boiling is thought to have occurred during Stage 1 in the Jinxiantou and Changgou deposits. This is supported by the following observations. (1) Numerous V-type FIAs occurred in Stage 1, indicating a large volume vapor phase during fluid trapping. (2)

The coexisting V and S-type inclusions homogenized at similar temperatures, which indicates that they were trapped at the same time and represent a boiling FIA. The partitioning of fluid into vapor-rich high-salinity fluids was caused by the boiling of parental magmatic fluids (Figure 10). (3) Most of the S-type FIs were homogenized by the simultaneous disappearance of the vapor phase and daughter minerals [39].

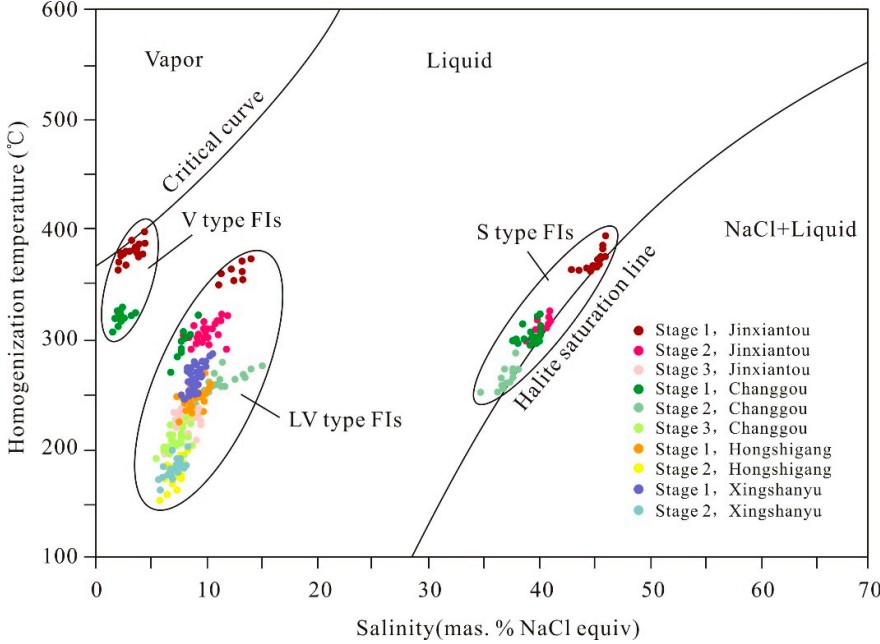

**Figure 10.** Plot of Th vs. salinity for fluid inclusions in quartz representing the different mineralization stages of the four deposits in the Qibaoshan polymetallic ore field.

The homogenization temperatures of the boiling inclusions were close to the quartz crystallization temperatures, which were similar to the ore formation temperatures of the Jinxiantou and Changgou deposits. The trapping pressures could be estimated using the lowest homogenization temperatures and salinities of the boiling FIA (V and S type) in Stage 1 and the isobar equations of Hedenquist et al. [40] and Redmond et al. [41]. However, the S-type FIs that homogenized during halite disappearance are excluded from the calculation, because they could not have been trapped from boiling fluids [42,43] and may have re-equilibrated at higher pressures.

As shown in Figure 11, the estimated trapping pressures of the coexisting Stage 1 V and S-type inclusions are 150 bar and 80 bar for the Jinxiantou and Changgou deposits, respectively. The mineralization zones have clear boundaries with the surrounding rocks, indicating that mineralization occurred under hydrostatic pressure [44,45]. Based on these results, the mineralization in the Jinxiantou and Changgou deposits is inferred to have occurred at depths of ~1.5 km and ~0.8 km assuming hydrostatic pressure, respectively.

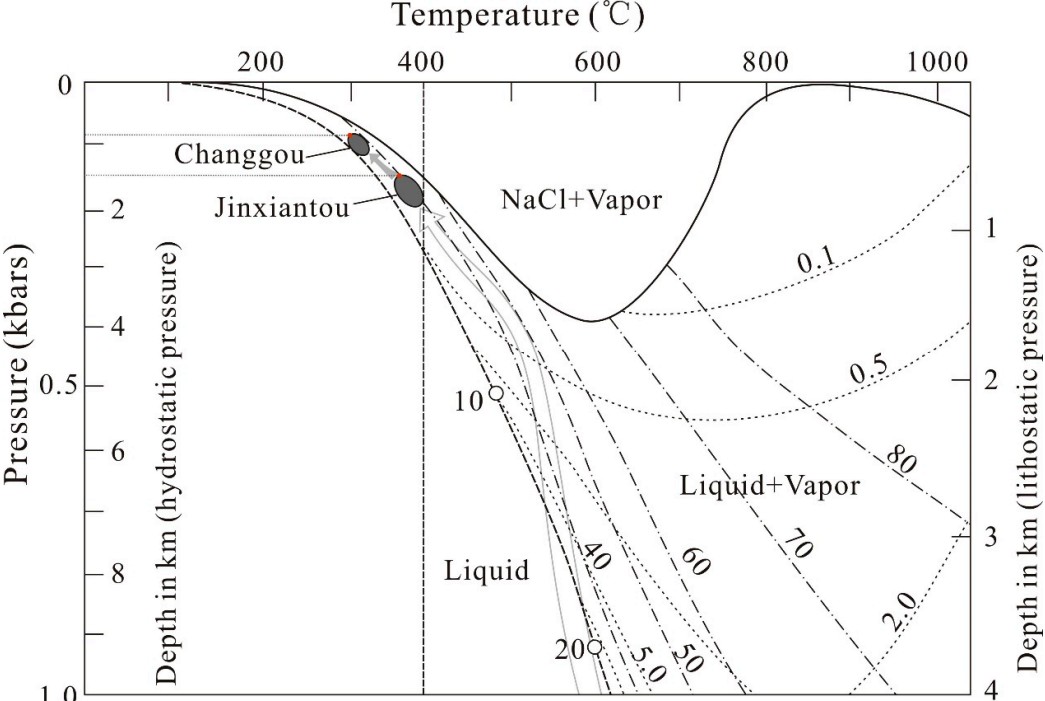

**Figure 11.** Pressure–temperature (P–T) diagram for the H$_2$O–NaCl system in the Changgou and Jinxiantou deposit (adapted from Redmond et al. [41], Fournier [46], and Hedenquist et al. [44]). The brittle ductile transition zone lies at about 400 °C [46]. The brittle plastic transition zone lies at about 400 °C [44]. The critical curve is shown by the dashed thick line with labels for 10 mas. %, 20 mas. %, and 30 mas. % NaCl equivalent. fluids. Dot dashed lines are contours of constant mas. % NaCl dissolved in brine (boiling-point curves) with values for 40, 50, 60, 70 and 80 wt. % NaCl equivalent. Dotted lines are contours of constant mas. percent NaCl dissolved in vapor (condensation curves) with values for 0.1 mas. %, 0.5 mas. %, 2.0 mas. %, and 5.0 mas. % NaCl equivalent, respectively.

With continued mineralization in the Jinxiantou and Changgou deposits, the number of V and S-type FIs gradually decreased, while the number of LV-type FIs gradually increased. The ore-forming fluid changed from a medium-temperature, high-salinity boiling system to a low-temperature, low-salinity homogeneous system, which may have been the result of fluid mixing.

In the Xingshanyu and Hongshigang deposits, the inclusions contain a fluid with a complex composition and T$_e$ values ranging from −34.8 to −28.6 °C, which is typical of Na, K, (Mg)-chloride solutions [27]. Only the LV-type FIs are present within the quartz granules. From the early to late mineralization stages, the salinity and homogenization temperature of the ore-forming fluid gradually decreases. The mineralization occurred in a low temperature, low salinity homogeneous fluid system. The trapping pressures could not be estimated due to a lack of evidence of fluid boiling.

### 7.3. Sources and Evolution of the Hydrothermal System

The possibility of different fluids with different origins being present within the hydrothermal system can be addressed using the H and O isotopic data (Table 2, Figure 12).

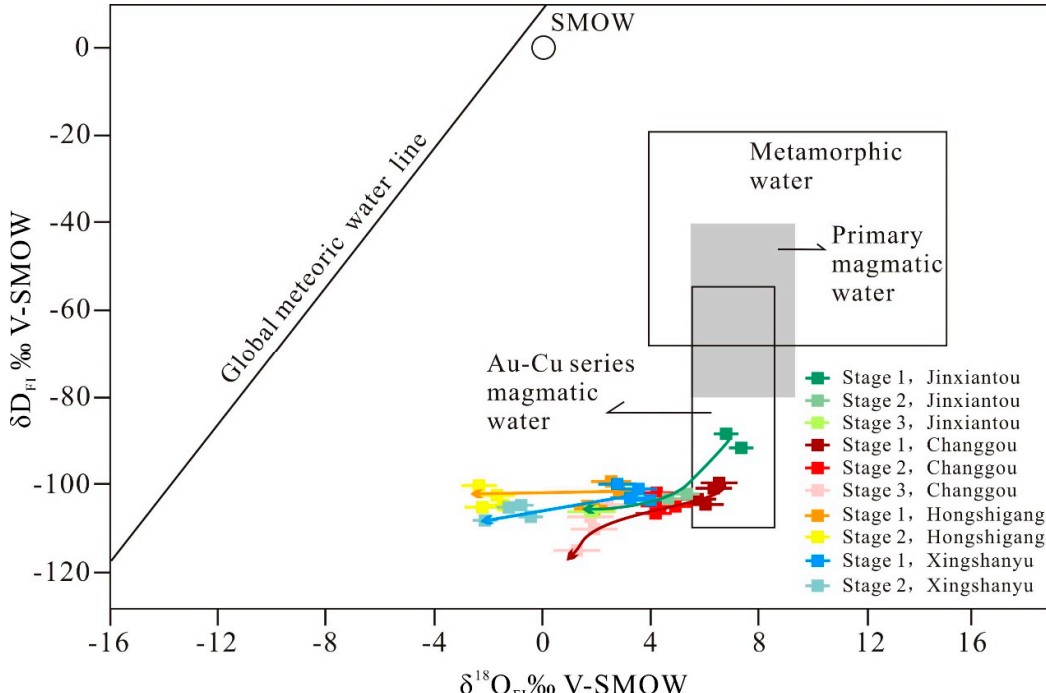

**Figure 12.** Plot of δD vs. δ¹⁸O for fluid inclusions in quartz representing the different mineralization stages of the four deposits in the Qibaoshan polymetallic ore field. The fields for metamorphic, primary magmatic waters and global meteoric water line are from Taylor [32]. The field for Au–Cu series magmatic water is from Zhang [45].

The hydrogen and oxygen isotopic compositions of the Jinxiantou and Changgou deposits are similar.

On the δD versus δ18O plot, the δ18O values of the Stage 1 samples plot in the primary magmatic water field, while the δD values are 10‰ to 25‰ lighter than the lightest primary magmatic water.

The lighter δD values can be produced by several methods: (1) rock–water interactions, (2) hydrogen isotope fractionation, (3) fluid boiling, (4) ore fluids that have evolved by mixing with various types of water, e.g., local meteoric water, and (5) magmatic degassing.

Next, we assessed each of these methods. Method 1: Minerals such as the biotite and hornblende common in volcanic complexes can have δD values of −170‰ [47]. As the ore-forming fluid flows through the wall rock, high-T water–rock interaction could lead to isotopic exchange, resulting in a decrease in the δD of the evolved fluid. Method 2: The lighter δD values may also be due to hydrogen isotopic fractionation between water and a reduced species such as $CH_4$ [32]. However, no methane-containing FIs were found, and this effect is limited in hydrothermal systems due to the large volume of $H_2O$ [32]. Method 3: Boiling the fluids can influence the hydrogen isotopic composition because vapor separation results in the relative depletion of D in the remaining ore-forming fluid [48]. Boiling of the fluid was confirmed by fluid inclusion microthermometry in the Jinxiantou and Changgou deposits. Method 4: The mixing of magmatic water and the heating of meteoric groundwater may be a plausible explanation for the observed low δD values. These low δD values were potentially caused by the incorporation of local meteoric water. However, the input of meteoric water often leads to the decrease of the δD and δO values of the fluid at the same time. No obvious shift of stage 1 δO value indicated that the ore-forming fluid was dominantly magmatic water, with little or no addition of meteoric water. Method 5: Magmatic degassing can produce significant ranges of δD and δ34S at the same time through fractionation. Different degrees of degassing in an open system could cause the δD of ore-forming fluids derived from magmatic water

to be depleted by 50–80% [49]. However, the δ³⁴S data for the Jinxiantou and Changgou deposits has a limited range, indicating that fractionation did not occur.

The Stage 1 samples plot in the Au–Cu series magmatic water field, as defined by Zhang [45]. The characteristics of isotopic in Jinxiantou and Changgou are similar to many Au–Cu deposits in China, as reported by Zhang [45].

The O–H isotopic data for the paragenetically younger Stage 2 quartz samples defined a trend toward the meteoric water line. Compared with Stage 1, V-type FIs are absent in the Stage 2 samples, while the number of S-type FIs decreases and the number of LV-type FIs increase. These characteristics may indicate the beginning of fluid mixing during mineralization. The Stage 3 samples plot closer to the meteoric water line, and only the LV-type FIs are present in this stage. These characteristics could be the result of increasing contributions of meteoric water with time.

In conclusion, the ore-forming fluid in Jinxiantou and Changgou deposits was derived from a magmatic source that underwent intense water–rock interaction and fluid boiling and lighter hydrogen isotope values. Then, the ore fluid mixed with meteoric water. This mixing became more pronounced in the hydrothermal system during the late mineralization stage. All of the data for the Hongshigang and Xingshanyu deposits lie on a horizontal trend, to the right of the meteoric water line. Furthermore, the trend does not point to the "primary magmatic water" box. Only LV-type FIs were observed in the quartz granules, and the temperature () and salinity of the fluid were 155–289 °C and 5.6–10.5 mas. % NaCl, respectively. This indicates that the ore-forming fluids most likely originated from circulating meteoric fluids with little or no addition of magmatic water. The ore-forming fluids reacted with wall rocks and extracted ore-forming elements, so the salinity of ore-forming fluids is obviously higher than that of meteoric water. In the Hongshigang and Xingshanyu deposits, the initial ore-forming fluid was dominated by circulating meteoric water. Meteoric water generally has low δD and δ¹⁸O values [32,50]. On the δD versus δ¹⁸O diagram, the data points form a relatively straight line with only a small variation in δ¹⁸O (oxygen isotope shift). Driven by a hot magma source, the meteoric water constantly exchanged oxygen isotopes with the surrounding silicate and carbonate containing rocks at high temperatures. The size of the oxygen isotopic shift reflected the degree of isotopic exchange.

In conclusion, the ore-forming fluids in the Hongshigang and Xingshanyu deposits were derived from circulating meteoric water that underwent isotopic exchange with the surrounding rocks.

*7.4. Two Phases of Mineralization in the Qibaoshan Polymetallic Ore Field*

Numerous studies have been conducted on the polymetallic ore field [1–5]. These studies increased our understanding of mineralization processes. However, the relationship between the Pb–Zn mineralization and the Cu (–Au) mineralization is still unclear. In the following discussion, we make two assumptions.

1. Development of the porphyry-hydrothermal vein system involved two types of mineralization. The mineralization center is in the Jinxiantou area. The formation of this Pb–Zn mineralization is the result of the outward migration of ore-forming fluids and elements.

2. Two phases of mineralization occurred in the Qibaoshan area: Au–Cu mineralization and Pb–Zn mineralization.

The four deposits in the Qibaoshan ore field are different in mineralization ages, fluid characteristics, and mineral assemblages.

The Jinxiantou deposit is believed to be a quartz diorite porphyry deposit [1,2]. The quartz diorite porphyry yielded a U–Pb age of 125 Ma [23], which may approximately represent the age of Au–Cu mineralization. The ore body in the Xingshanyu deposit crosscuts the diorite, which yielded a U–Pb age of 112 Ma [23]. The age of the diorite represents the upper limit of the age of the Pb–Zn mineralization. We suggest that the ore field has undergone at least two phases of mineralization.

The initial ore-forming fluids of the Jinxiantou and Changgou deposits had temperatures of 286–397°C and salinities of 36.7–45.8 mas. % NaCl, and were derived from magmatic water. The

ore-forming fluid was a Na-chloride solution that experienced significant boiling. The Cu (–Au) mineralization was caused by the emplacement of the quartz diorite porphyry. The ore-forming fluids of the Hongshigang and Xingshanyu deposits had temperatures of 155–289 °C and salinities of 5.6–10.5 mas. % NaCl, and were derived from meteoric water. The ore-forming fluid was typical of Na, K-chloride solutions that did not undergo fluid boiling. The Pb–Zn mineralization was likely caused by a concealed intrusion, but the concealed intrusion only heated the meteoric water. The mineral assemblage of the Jinxiantou deposit is dominated by specularite, chalcopyrite, and native gold, while the Hongshigang and Xingshanyu deposits are dominated by galena and sphalerite. The Changgou deposit is unique because it contains Cu–Pb–Zn mineralization. Significant amounts of specularite also occurred in the Changgou deposit, whereas it is absent in the Hongshigang and Xingshanyu deposits. We conclude that these characteristics are the result of late Pb–Zn mineralization superimposed on early Cu (–Au) mineralization.

Therefore, the most reasonable explanation is that the Qibaoshan area experienced two phases of mineralization. The Au–Cu mineralization event occurred at ~125 Ma, and the Pb–Zn mineralization event occurred after 112 Ma. We speculate that the early Au–Cu mineralization center is located in the Jinxiantou area. The Cu (–Au) ore-forming fluid flowed from the Jinxiantou area to the Changgou area (Figure 13). The late Pb–Zn mineralization center is located in the Xingshanyu area. The Pb–Zn ore-forming fluid flowed from the Xingshanyu area, through the Hongshigang area, and into Changgou the area. The two phases of ore-forming hydrothermal fluids are superimposed on the Changgou deposit, creating unique ore-forming conditions.

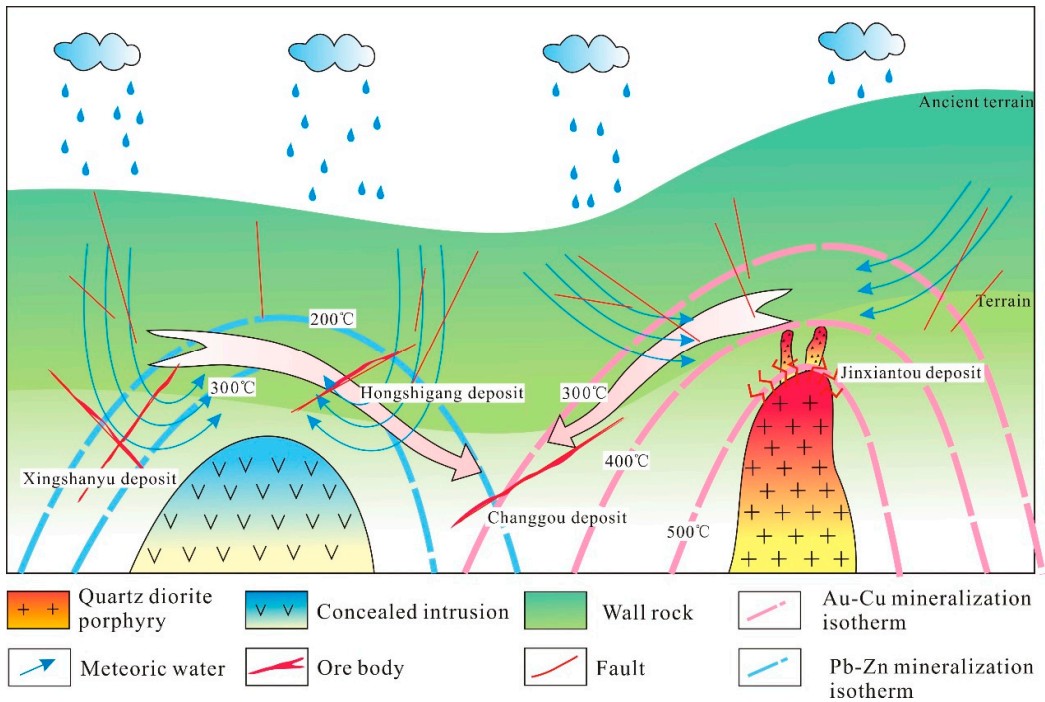

**Figure 13.** Metallogenic model for the Qibaoshan polymetallic ore field.

## 8. Conclusions

(1) The primary source of sulfur for the four deposits in the Qibaoshan ore field was magmatic with minor wall rock contamination.

(2) The ore-forming fluids of the Jinxiantou and Changgou deposits were derived from a magmatic source that underwent intense water–rock interaction followed by mixing with meteoric water. The ore-forming fluids of the Hongshigang and Xingshanyu deposits were derived from circulating meteoric water that underwent isotopic exchange with the surrounding rocks.

(3) Two phases of hydrothermal activity occurred in the Qibaoshan polymetallic ore field. Late Pb–Zn mineralization was superimposed on early Cu (–Au) mineralizaton in the Changgou deposit.

**Author Contributions:** Conceptualization, S.-D.L. and G.-Y.Y.; Methodology, K.-Y.W.; Software, Y.-C.W.; Validation, K.-Y.W.; Formal Analysis, Y.-C.W.; Investigation, S.-D.L.; Resources, K.-Y.W.; Data Curation, S.-D.L. and G.-Y.Y.; Writing-Original Draft Preparation, S.-D.L. and G.-Y.Y.; Writing-Review & Editing, S.-D.L. and K.-Y.W.; Visualization, S.-D.L.; Supervision, Y.-C.W.; Project Administration, S.-D.L. and K.-Y.W.; Funding Acquisition, K.-Y.W.

**Funding:** This work was supported by the Shandong Gold Group (Project No. KY201307).

**Acknowledgments:** We are grateful to the staff of the Shandong Gold Group for assisting with field work in the Qibaoshan Mine. We are grateful to Hanbin Liu for his assistance with the stable isotope analyses at the Analytical Laboratory of the Beijing Research Institute of Uranium Geology. We would also like to thank the managing editor and reviewers for their positive and constructive comments, which significantly improved this paper.

**Conflicts of Interest:** The authors declare that there is no conflict of interest regarding the publication of this paper.

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
