# Peer review of "Fluid Evolution and Ore Genesis of the Qibaoshan Polymetallic Ore Field, Shandong Province, China: Constraints from Fluid Inclusions and H–O–S Isotopic Compositions"

_minerals, doi:10.3390/min9070394_

Reviewer 1 Report

I found the paper well written, the analytical methods adequate, the discussion mature and the conclusions well supported by the data; there are some minor errors and misprints in the text and some inaccuracies in the references. The histogram boxes in figure 8 must be renamed. See the attached file.

Author Response

Respected reviewer:

I appreciate your time in reviewing my manuscript so soon. Your opinion is very important and precious to me. The paper has been carefully thinking and modified. There are some detailed changes as follows:

1. Line 26

Modify “mixied” to “mixed”.

2. Line 29

Modify “quatz” to “quzrtz”.

3. Line 35

Modify “Xingshenyu” to “Xingshanyu”.

4. Line 84-87

Modify “strata” to “rocks”.

5. Line 119

Modify “sericitization” to “sericitized”.

6. Line 154-159

Modify “poor sulfide” to “sulfide-poor”.

7. Line 224

Modify “quatz” to “quzrtz”.

Modify “poor sulfide” to “sulfide-poor”.

8. Renamed the histogram boxes in figure 8.

9. Line 353

Modify “valuesof” to “values of”.

10. Line 408

Modify “minerliazion” to “mineralization”.

11. Line 411

Modify “wee” to “we”.

12. Line 441

Modify “plastic” to “ductile”.

13. Line 530

Modify “natural” to “native”.

14. Modify the references.

Thanks again for your guidance and valuable suggestions. If you have any queries, please don’t hesitate to contact me through Manuscript Tracking System.

Best regards.

Shunda Li

Reviewer 2 Report

Dear authors,

you will find many specific comments in the annotated manuscript. Here I discuss a few more general issues and suggestions. 

The CO2 issue

In Introduction the authors write: “Wang et al. [1] suggest that the composition of the gas phase of the fluid inclusions (FIs) is mainly CO2, and the ore-forming temperature is 150–320°C. The gold mineralization is closely related to the CO2 content“ (the quote refers to Jinxiantou Au-Cu deposit) The alleged presence of CO2 in fluid inclusions represents MAJOR disagreement with the results presented in this paper: the authors do not report CO2 from fluid inclusions at this deposit. This disagreement ABSOLUTELY must be discussed and somehow explained. Such discussion is presently missing from the text (in fact, the discrepancy not even mentioned).

Nomenclature

Use the form: mas.% NaCl-equiv. (the form "wt.% "is obsolete and its use must be discouraged)

DO NOT use form d18OV-SMOW. Lower index indicated WHAT was measured – not the normalization scale. The latter is conventionally reported after the ‰-sign. Example: d18Oqz = 7.2 ‰ V-SMOW. Same convention applies to dD and d34S. Since only two normalization scales are used through the paper, it is advisable to define them in the Methods sections (something like: “All d18O and dD values reported in this paper are normalized vs. V-SMOW; the d34S values – vs. V-CDT scales”). And then simply report the values in permil.

Use d18Oqz to denote values obtained from quartz, d18OH2O and dDH2O (or d18OFI and dDFI) – for values pertaining to fluid inclusions.

V or VL-type inclusions?

In the text – vapor rich inclusions are denoted as V type. In Table 1, which summarizes the FI data there is no V-type, but the VL-type is present instead. The authors must make presentation consistent.

Fluid inclusions micro-thermometry data reporting and analysis

Histogram in Fig. 8 is a good way of presenting the data. They are not as useful in data analysis. In this regard, diagrams Th vs. Salinity (showing also critical curve H2O-NaCl and halite saturation curve) is typically very useful. It may graphically show the phase separation (if present)

Table 2 - over-reporting

As we know, all samples return not a single Th but a range of measured Th’s. The number given in the Table is a mean Th, which must be reported along with its confidence interval (e.g., 320±15°C at 1s).

This is important, because the mean value is further used to calculate d18OH2O of FI water (last column). The uncertainty in Th must be propagated into the uncertainty in calculated d18OH2O values – the latter, too must be reported with uncertainty (e.g., 7.0±1.2‰ at 1s). Without his, numbers in the table convey a (wrong) sense of confidence; the data, in fact, are less precise than they appear.

Table 3 (S isotopes)

Detailed explanation must be provided for this table. Now many parameters reported in the table are not explained, and a reader will have difficulty analyzing the data (as I have).

What is d34SH2S(frequent)? Why it is needed to be presented?

What is d34SH2S(highest)? Why it is needed to be presented? I note that most of the values in this column are either identical or SMALLER than those reported in column “frequent” – how can that be?

Why model value (Th(mode)) is used here (column #7), while mean Th was used in Table 2? Rationale for changing the approach must be presented.

What is the meaning of Th (highest)? As a fluid inclusionists, I have no value in reporting this number – what is the rationale behind it?

Codes in the last column (Location) are not helpful, because they are not explained in the text. A reader has no clue what do they mean – so why provide them? Name this column “Depth from surface (m)” (and remove “m’s” from table body)

And, as in Table 2, lower index CDT in column #5 is not appropriate (ALL d34S values in this table are reported vs. this standard – why emphasize it in one column and omit in others?). Reporting values vrt. CDT standard must be indicated in the Table’s heading, and not used in the table.

Interpretation of stable isotope trends

The author’s discussion of the data presented in Fig. 11 is not satisfactory. They make rather arbitrary and judgmental statements, such as: “…the Stage 1 samples plot adjacent to the primary magmatic water field, indicating that the ore-forming fluid of the early mineralization stage was dominantly magmatic water, with little or no addition of meteoric water” (ll. 465-466). Problem is, in the Figure the data plot 10 to 25 ‰ off the “magmatic water box”.

They continue: “The Stage 3 samples plot closer to the meteoric water line, and only the LV FIs are present in this stage. These characteristics could be the result of increasing contributions of meteoric water with time” (470-472) and

All of the data for the Hongshigang and Xingshanyu deposits falls between the magmatic water and meteoric water fields.” (473-474)

The latest statement is, technically, wrong. The data for these deposits lie on a horizontal trend, to the right of the MWL. The trend does not point to the “magmatic water” box shown in the Figure. .

Mixing between "magmatic” and “meteoric" fluids must follow a tie-line between the end members.  In this case, however, isotopic shift is strictly to the left (hydrogen isotope values do not change). We have to assume, then that the magmatic water in this region has hydrogen isotope value of -110-120 ‰. Is that reasonable?

In my opinion, the author’s interpretation is not supported by the data they report. They must consider and discuss alternative explanations. What about high-T water-rock interaction (producing positive O shift)? The discussion must be accompanied by at least minimal numeric calculations of possible isotope effects.

Importantly, in any interpretation involving meteoric waters they must explain the high salinities of hose (which are, even at late stages are significantly higher than sea water salinity).

Author Response

Respected reviewer:

I appreciate your time in reviewing my manuscript so soon. Your opinion is very important and precious to me. The paper has been carefully thinking and modified. There are some detailed changes as follows:

1. The CO2 issue

It’s my mistake. I modified the sentences and made some explanation in 6.1 section .(Line 274-276, clean manuscript)

Wang et al. [1] only report that the composition of the gas phase of the fluid inclusions (FIs) contain CO2 (not CO2 FIS). The lack of liquid CO2 or clathrate formation during freezing indicates that none of the inclusions contained significant quantities of CO2. The content of CO2 is less in gas phase composition (16.7 μL/g) as reported by Wang et al [1].

2. Nomenclature

Unified and revised the nomenclature throughout the manuscript.

mas. % NaCl equivalent to denote the salinity of FIs

δ18Oqz to denote values obtained from quartz.

δ18OFI to denote values obtained from fluid inclusions.

δ34Ssulfide to denote values obtained from sulfide.

δ34SH2S to denote values obtained from H2S.

Define the normalization scales in the Methods sections.

Vienna Standard Mean Ocean Water (VSMOW) and Vienna-Canyon Diablo Troilite (VCDT).

3. V or VL-type inclusions

Revised and unified the type of fluid inclusions throughout the manuscript.

LV type:liquid-rich aqueous inclusions

V type:vapor-rich aqueous inclusions

S type:halite-bearing inclusions

4. Fluid inclusions micro-thermometry data reporting and analysis

Add new Th vs. Salinity illustration. (Figure 10).

We selected regular inclusions to estimate the volume ratio.

5. Table 2 - over-reporting

Reported the mean Th and calculated δ18OFI values along with confidence interval.

6. Table 3 (S isotopes)

Used the mean Th with confidence interval to calculate the δ34OFI values.

7. Interpretation of stable isotope trends

We discussed the causes of lighter H isotope and considered the effects of high-T water-rock interaction. (Line 484-507, clean manuscript)

“The lighter δD values can be produced by several methods: 1) rock-water interactions, 2) hydrogen isotope fractionation, 3) fluid boiling, 4) ore fluids that have evolved by mixing with various types of water, e.g., local meteoric water, and 5) magmatic degassing…….”

“The ore-forming fluids reacted with wall rocks and extracted ore-forming elements, so the salinity of ore-forming fluids is obviously higher than that of meteoric water.”

8. Method

In the test method (5.2 section), the decrepitation T, repeatability, calculation method (δ18OFI18Oqz -1000lnαqz-water), and measurements calibrated method are improved. (Line 250-262, clean manuscript)

It is not a vacuum system of continuous-flow. The H2 was collected by sample collection tube and then transferred to the mass spectrometer for analysis.

9. Sources of ore constituents

Thanks for your advice. This section was re-described. And add more information to the isotope analysis. (Line 396-421, clean manuscript)

10. Modified the spelling mistakes, repeated and inaccurate sentences.

11.Figures and Tables

Revised the figures and illustrations according to the comments

At the bottom of the tables, necessary explanation of terms and calculation formula are added.

12. References

Add needed references and rearrange the order of references

The revisions are detailed in the manuscript.

Thanks again for your guidance and valuable suggestions. If you have any queries, please don’t hesitate to contact me through Manuscript Tracking System.

Best regards.

Shunda Li

Round  2

Reviewer 2 Report

Dear authors,

I am satisfied with the way you implemented my suggestions for paper imprvement.

Please - do a careful reading of you text and correct small English errors (singular-plural, etc.).

Good luck!